# CONSUMERBENCH: Benchmarking Generative AI Applications on End-User Devices

## ABSTRACT

The recent shift in Generative AI (GenAI) applications from cloud-only environments to end-user devices introduces new challenges in resource management, system efficiency, and user experience. This paper presents CONSUMERBENCH, a comprehensive benchmarking framework designed to evaluate the system efficiency and response time of GenAI models running on end-user devices. Unlike existing benchmarks that assume exclusive model access on dedicated GPUs, CONSUMERBENCH simulates realistic multi-application scenarios executing concurrently on constrained hardware. Furthermore, CONSUMERBENCH supports customizable workflows that simulate complex tasks requiring coordination among multiple applications. CONSUMERBENCH captures both application-level metrics, including latency and Service Level Objective (SLO) attainment, and system-level metrics like CPU/GPU utilization and memory bandwidth. Through extensive experiments, CONSUMERBENCH reveals inefficiencies in resource sharing, unfair scheduling under greedy allocation, and performance pitfalls of static model server configurations. The paper also provides practical insights for model developers and system designers, highlighting the benefits of custom kernels tailored to consumer-grade GPU architectures and the value of implementing SLO-aware scheduling strategies.

## 1 INTRODUCTION

The deployment landscape for Generative AI (GenAI) models is undergoing a significant transformation: once confined to hyperscale data centers equipped with powerful GPUs, these models are now increasingly being adopted on local devices such as laptops and smartphones Android Developers (2025); Apple Inc. (2024a); Qualcomm Technologies, Inc. (2023). This shift is primarily motivated by growing concerns over data privacy, latency, and availability under various network conditions.

Beyond simple resource constraints, end-user devices present unique challenges due to their heterogeneous application landscapes that compete for the limited hardware resources. Each application often requires specific types of GenAI models due to model performance considerations Hajikhani & Cole (2024); Shi et al. (2023) and varies significantly in its runtime characteristics. For example, some applications (*e.g.*, deep research agents or image generation) are long-running background processes, whereas others (*e.g.*, chatbots or live audio captioning) are short-lived tasks demanding instant response. Each of these categories imposes distinct Service Level Objectives (SLOs). Unlike cloud environments, where the different models and applications can be distributed across separate servers and dedicated GPUs, end-user devices must accommodate all models on a single, shared GPU. Consequently, end-user devices must be capable of concurrently executing the diverse models, effectively managing the limited compute, memory, network, and power resources, while consistently meeting diverse SLOs.

Existing research on GenAI inference for end-user devices Dettmers et al. (2023); Frantar et al. (2023); Lin et al. (2024) and specialized inference frameworks Gerganov & ggml-org contributors (2023) aim to achieve target quality, latency, and throughput within platform constraints. However, these approaches typically assume that the model has dedicated, exclusive access to the hardware. Furthermore, existing benchmarks that evaluate the computational efficiency of GenAI models often assume that hardware resources are dedicated to a single application Reddi et al. (2020); Li et al. (2025); Jayanth et al. (2024); Laskaridis et al. (2024); Xiao et al. (2024), failing to accurately represent the end-user experience when running multiple models on their devices. As a result, the complexities and opportunities associated with efficiently deploying these models in local environments, characterized by other concurrently running applications, remain unexplored.

To address this gap, we present CONSUMERBENCH, a comprehensive benchmarking framework that evaluates the runtime performance of user-defined GenAI applications under realistic conditions on end-user devices. Users specify the GenAI applications, models, request patterns, and SLOs (*e.g.*, latency, throughput) in a simple configuration file. CONSUMERBENCH then evaluates application performance against these SLOs across diverse deployment scenarios, including GPU/CPU hybrid setups, resource partitioning, and shared model deployments (*e.g.*, inference servers hosting models for multiple applications). Beyond core performance metrics, it captures system-level data such as GPU/CPU utilization, memory bandwidth, and power consumption to quantify the efficiency of on-device GenAI execution. CONSUMERBENCH is also the first to benchmark user-defined collaborative workflows, such as "creating a YouTube video," where multiple GenAI applications (*e.g.*, text-to-image, speech recognition, thumbnail generation) interoperate to complete complex tasks. This enables realistic testing of how resource orchestration impacts end-to-end user experience.

Through the evaluation of a diverse set of GenAI applications on a local server with a consumer-grade GPU, CONSUMERBENCH offers novel insights for the development of efficient infrastructure tailored to support end-user devices. CONSUMERBENCH reveals that greedy GPU resource allocation leads to severe starvation of lightweight applications (*e.g.*, live captioning), while static GPU partitioning reduces overall throughput due to resource underutilization. Furthermore, inference server-based model sharing fails to meet diverse application SLOs when configurations are not tailored, highlighting the need for dynamic, SLO-aware memory management and scheduling strategies as well as GPU architecture-aware kernel designs. In summary, this paper makes the following contributions:

- It highlights unique challenges in running diverse GenAI applications concurrently on end-user devices, including resource contention and unmet SLOs, that are not exposed by prior benchmarks.

- It introduces a benchmarking framework called CONSUMERBENCH that supports realistic multi-application workflows, tracks both application- and system-level metrics, and allows flexible configurations for evaluation on end-user devices.

- It evaluates four representative applications, reveals key system inefficiencies under different GPU sharing strategies, and offers unique insights into building efficient systems for end-user devices.

## 2 RELATED WORKS

### 2.1 LLMS ON END-USER DEVICES

Cloud inference has long been the default for GenAI applications, but recent advances in resource-efficient models Abdin et al. (2024); AI (2024); Yang et al. (2024), model compression Dettmers et al. (2023); Frantar et al. (2023); Lin et al. (2024), efficient runtimes Gerganov & ggml-org contributors (2023) and edge-oriented hardware Apple Inc. (2024a); Qualcomm Technologies, Inc. (2023); Android Developers (2025) have made credible *on-device* inference possible.

The effects of executing multiple heterogeneous GenAI applications concurrently on a single device, however, remain mostly unexplored. Prior work Mei et al. (2024); Packer et al. (2023) has built OS-like abstraction for GenAI applications, but they focus on single application rather than tracking application-level SLOs under interference. Others have studied co-locating workloads on the GPU Yu et al. (2024); Strati et al. (2024); Han et al. (2022), but they either focus on distributed clusters, restrict to a single workload type such as DNN inference, or require modifications at the application level.

### 2.2 BENCHMARKS FOR EDGE LLM INFERENCE

Existing benchmarks for edge-class hardware, including MLPerf Inference Reddi et al. (2020), PalmBench Li et al. (2025), MELT Laskaridis et al. (2024), and others Xiao et al. (2024); Jayanth et al. (2024), primarily assess latency, throughput, and energy for single-stream inference, often with exclusive hardware access. Other benchmarks Liu et al. (2023); Guo et al. (2024); Li et al. (2023a); Xu et al. (2024); Deng et al. (2024); Murthy et al. (2024) evaluate model capabilities on local/mobile systems but lack evaluation of system performance. In contrast, CONSUMERBENCH benchmarks the concurrent execution of *heterogeneous* workflows comprising multiple GenAI tasks, thereby highlighting cross-application interference patterns and system inefficiencies.

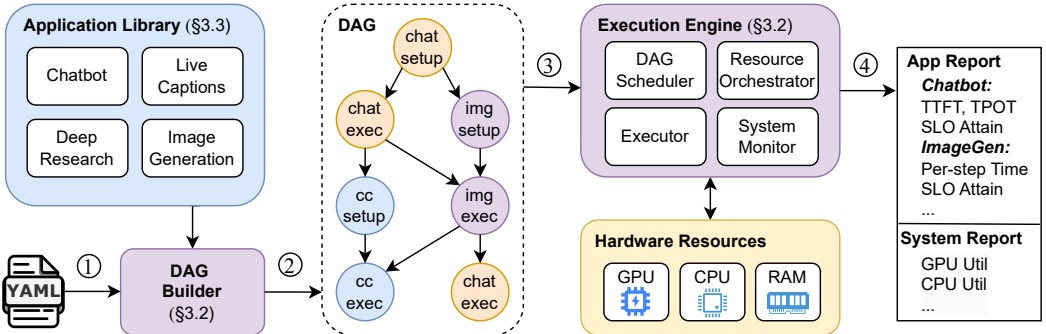

Figure 1: The overall design of CONSUMERBENCH.

```
1  Analysis (DeepResearch):
2     model: Llama-3.2-3B
3     num_requests: 1
4     device: cpu
5  Creating Cover Art (ImageGen):
6     model: SD-3.5-Medium-Turbo
7     num_requests: 5
8     device: gpu
9     slo: 1s
10 Generating Captions (LiveCaptions):
11    model: Whisper-Large-V3-Turbo
12    num_requests: 1
13    device: gpu
14    ...
```

(a) Task Definition

```
1  analysis_1:
2     uses: Analysis
3  cover_art:
4     uses: Creating Cover Art
5     depend_on: ["analysis_1"]
6  analysis_2:
7     uses: Analysis
8     depend_on: ["analysis_1"]
9  generate_captions:
10    uses: Generating Captions
11    depend_on: ["cover_art",
12    "analysis_2"]
13    ...
14
```

(b) Workflow Definition

Figure 2: Example YAML configuration to define application tasks as well as user workflows.

## 3 CONSUMERBENCH

CONSUMERBENCH is a comprehensive benchmarking framework specifically designed to evaluate the runtime performance of GenAI models on end-user devices under realistic conditions.

### 3.1 GOALS

- **G1** : **Diversity of Applications.** CONSUMERBENCH supports various GenAI applications. They range from long-running background services to short-lived, latency-sensitive interactive tasks with tight SLOs, and with different modalities (*e.g.*, text-to-text, text-to-image). See §3.3.

- **G2** : **Concurrent Execution and Resource Contention.** CONSUMERBENCH evaluates performance and SLO attainment when multiple applications run concurrently on shared limited hardware and use different resource orchestration strategies. This reveals resource contention and trade-offs overlooked by existing benchmarks. See §3.2.

- **G3** : **System-Level Holistic Metrics.** CONSUMERBENCH collects detailed metrics such as compute utilization, memory bandwidth utilization, and power consumption. These metrics help users not only understand whether an application meets its SLO, but also diagnose why it might fail, which is crucial for improving system design and optimizing on-device deployment strategies. See §3.2.

- **G4** : **Configurability and Automation.** CONSUMERBENCH allows users to easily define application scenarios and performance goals through a user-friendly configuration. It automates execution and metric collection for complex workflows. See §3.2.

### 3.2 METHODOLOGY

Fig. 1 shows the design overview of CONSUMERBENCH. Given a user configuration of the workflow, CONSUMERBENCH orchestrates the execution of the workflow through a graph representation. After the workflow finishes, it generates a benchmark report for SLO satisfaction and resource efficiency.

**Input User Configuration (①).** The input to CONSUMERBENCH is a YAML configuration file that contains the set of applications to run, their corresponding models, hardware placements (CPU, GPU, or CPU-GPU hybrid), SLOs, and input requests. The configuration can also specify dependencies between applications for complex multi-step workflows. Users could either create workflows from scratch or from realistic GenAI application traces that record inference calls made during an execution. Fig. 2 provides an example YAML configuration for three applications (Deep Research agent, Image Generation and Audio Captioning), illustrating dependencies and SLOs. While our SLOs focus on request latency, users can configure SLOs for other metrics.

**Creating Overall Workflow (②).** CONSUMERBENCH builds a directed acyclic graph (DAG) from the YAML specification. Each node represents an application instance, with edges denoting dependencies. Nodes are of three types: `setup` (application startup), `exec` (execution), and `cleanup` (resource release). CONSUMERBENCH validates the DAG to ensure that there are no cycles and that each application includes a `setup` node before any `exec` nodes. Fig. 1 includes an example DAG with three applications: Chatbot (chat), ImageGen (img), and LiveCaptions (cc).

**Executing the Workflow (③).** The execution engine uses the DAG to coordinate application execution and collect metrics.

- The **DAG scheduler** manages request scheduling, respects dependencies, and enables concurrent execution where possible, supporting complex and realistic user-defined workflows.
- The **resource orchestrator** employs various GPU management strategies to execute workflows, including greedy resource allocation, where the applications consume GPU resources as needed, and GPU partitioning, which divides GPU resources equally among running applications.
- The **executor** is responsible for loading and unloading of the GenAI models and executing user requests, obeying the DAG scheduler.
- The **system monitor** tracks system-wide resource usage during execution. GPU compute and memory bandwidth usage are monitored with the help of NVIDIA's Data Center GPU Management (DCGM) utility Corporation (2025b).[1] CPU utilization is monitored using the `stat` tool, while CPU memory (DRAM) bandwidth utilization is monitored using the `pcm-memory` utility Corporation (2025a). CONSUMERBENCH monitors power consumption using NVML NVIDIA Corporation (2025) for GPU and RAPL Intel Corporation (2022) for CPU.

**Generating Benchmark Report (④).** After completing all workflow tasks, CONSUMERBENCH automatically evaluates each application's performance against the SLOs defined in the YAML configuration. It then generates a comprehensive report summarizing performance, SLO satisfaction, and resource efficiency for the entire workflow.

## 3.3 SUPPORTING DIVERSE APPLICATIONS AND MODELS

CONSUMERBENCH provides an API that enables users to integrate their own applications with either custom or existing models, making the framework highly extensible. To evaluate a custom application, users simply implement three functions: `setup()`, to initialize the GenAI application and its model; `execute()`, to send requests to the model; and `cleanup()`, to release application resources. CONSUMERBENCH further enables multiple applications to share a single model by using a common `setup()` function, which launches a shared inference server for all participating applications. This capability is important for end-user devices as it helps minimize memory usage and startup overhead. Table 1 summarizes the four applications currently supported by CONSUMERBENCH, covering different modalities and common use cases on end-user devices.

| Application | Dataset | Model |
|---|---|---|
| Chatbot | LMSYS-Chat-1M Zheng et al. (2024) | Llama-3.2-3B AI (2024) |
| DeepResearch | HotpotQA Yang et al. (2018) | Llama-3.2-3B AI (2024) |
| ImageGen | COCO Captions Chen et al. (2015) | SD-3.5-Medium-Turbo Studios (2024) |
| LiveCaptions | Earnings-21 Del Rio et al. (2021) | Whisper-Large-V3-Turbo Radford et al. (2022) |

Table 1: Summary of dataset and model used in each application.

**Chatbot.** This is a text-to-text generation app for chat and Q&A, featuring a frontend and a local backend with an OpenAI-compatible API OpenAI (2020). The backend uses llama.cpp Gerganov

---
[1]Although designed for datacenter GPUs, DCGM supports consumer-grade GPUs, including all GeForce GPUs.

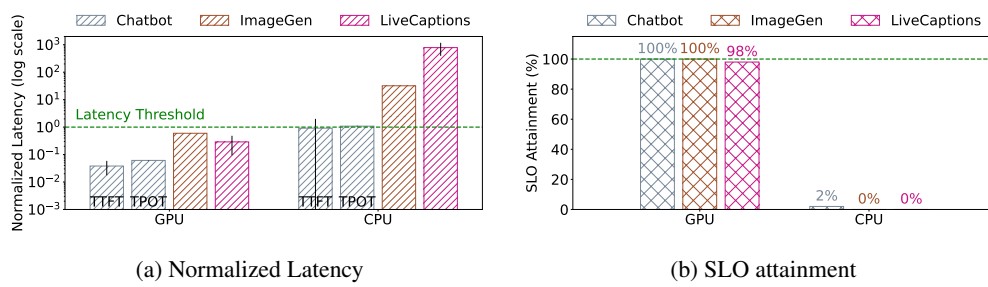

(a) Normalized Latency

(b) SLO attainment

Figure 3: (a) Latencies normalized to SLO requirements and (b) SLO attainment for Chatbot, Image Generation, and Live Caption running exclusively on the GPU or CPU.

& ggml-org contributors (2023), supporting CPU-GPU co-execution and is optimized for end-user devices. SLO targets are based on human reading speed Liu et al. (2024): 1 second for Time to First Token (TTFT) and a generation speed of 4 tokens per second (Time Per Output Token - TPOT).

**DeepResearch.** This is an agentic application for complex, multi-step reasoning and fact gathering, adapting smolagent's open-deep-research Roucher et al. (2025) and using LiteLLM BerriAI (2024) to interact with a local model via llama.cpp. DeepResearch operates on long contexts and functions as a persistent background application, without any SLO.

**ImageGen.** This is a text-to-image application with a simple frontend and a backend using stable-diffusion-webui AUTOMATIC1111 (2022) in API server mode. Motivated by research on diffusion models on resource-constrained devices Zhao et al. (2024); Li et al. (2023b), the SLO for ImageGen is set to 1 second per denoising step.

**LiveCaptions.** This is an audio-to-text application for real-time scenarios Macháček et al. (2023). The frontend chunks an audio file into segments and sends each segment to a backend adapted from whisper-online Macháček et al. (2023) to support HTTP connections. LiveCaptions can either be latency-sensitive or background. For the latency-sensitive case, a 2-second audio segment is sent every 2 seconds, requiring the model to generate captions in time. The SLO for this scenario is thus 2 seconds. For background transcription of a large audio file (5–10 minutes), there is no SLO.

## 4 EXPERIMENTATION

**Experimental Setup.** We run experiments on a local server consisting of a single RTX 6000 GPU NVIDIA Corporation (2018) with 24GB VRAM. The server is equipped with an Intel Xeon Gold 6126 CPU (2.60GHz, 24 cores) and 32GB of system memory (DRAM). The applications evaluated are listed in Table 1. Note that CONSUMERBENCH is not limited to the use of these applications, and users can add more applications by following the procedure in §3.3. For each application, CONSUMERBENCH samples requests from the dataset, measures per-request latency, and compares the results to the defined SLOs.

**System-wide metrics.** While CONSUMERBENCH monitors a range of resource utilization metrics such as compute, memory, and power (see §3.2), this section focuses on GPU compute (i.e., GPU utilization). Additional metrics are detailed in the Appendix. For GPU compute, CONSUMERBENCH tracks two key metrics: SMACT, the percentage of Streaming Multiprocessors (SMs) reserved by an application, and SM occupancy (SMOCC), the percentage of SMs actively running kernels. If SMOCC is significantly lower than SMACT, it indicates that the application is utilizing only a small portion of the SMs it reserves, suggesting inefficient use of GPU resources.

### 4.1 RUNNING APPLICATIONS EXCLUSIVELY ON GPU AND CPU

We establish performance bounds by running each application exclusively on the GPU (upper bound) and on the CPU (lower bound), to account for scenarios where the limited GPU memory may force applications to fall back to CPU execution. Fig. 3 shows results for our latency-sensitive applications; DeepResearch is excluded as it does not have an SLO. For Chatbot, we additionally evaluate the latest GPT-OSS-20B OpenAI et al. (2025) and find similar performance conclusions as Llama-3.2-

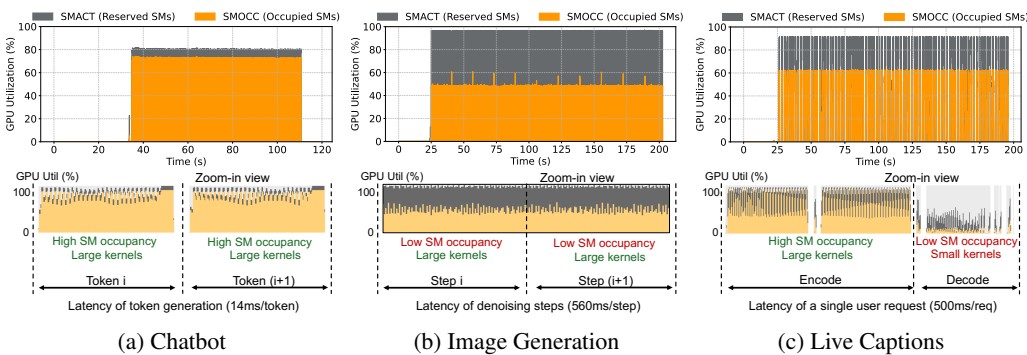

Figure 4: GPU utilization of each application running exclusively on the GPU.

3B Dubey et al. (2024) (see §A.3). Therefore, we will use Llama-3.2-3B Dubey et al. (2024) as the model for Chatbot and DeepResearch in the evaluation.

**Performance.** When running on the GPU, all applications achieve 100% SLO attainment except for LiveCaptions, where three[2] out of 150 audio segments incur SLO violations. When running on the CPU, all applications experience reduced performance, though the impact varies. Chatbot narrowly misses its SLOs, while ImageGen and LiveCaptions suffer from significantly higher request latencies.

**System-level metrics.** Fig. 4 shows the GPU utilization when applications run exclusively on GPU. All applications reserve almost all of the GPU cores when running exclusively, shown using SMACT. However, their efficiencies are very different, indicated by SMOCC. Chatbot utilizes its resources efficiently, while ImageGen and LiveCaptions under-utilize their reserved GPU cores.

**Analysis of Individual Applications.** We analyze the GPU kernels launched by each application to understand their performance and GPU utilization, shown through the zoomed-in view, in Fig. 4.

- **Chatbot:** Spends majority of its time in decoding output tokens (*i.e.*, token generation). It achieves a high SMOCC (Fig. 4a) since llama.cpp customizes the thread block and grid dimensions for its kernels during prefill and decode according to the underlying GPU architecture.
- **ImageGen:** Spends most of its time in the denoising phase, which uses an attention-based U-Net Oktay et al. (2018). PyTorch's generic attention kernel used by SD-3.5-Medium-Turbo requires over 150 registers per thread, limiting the number of threads that can run concurrently on each SM. This reduces SMOCC and leads to suboptimal GPU utilization (Fig. 4b).
- **LiveCaptions:** Uses the encoder-decoder Whisper-large-v3-turbo model. The encoder phase performs parallel operations on the input audio, involving softmax and large matrix multiplications that use tens of registers per thread with a high SMOCC. The decoder phase, however, spends most time in performing matrix multiplications with much smaller kernels than Chatbot or ImageGen. This phase also involves hundreds of registers per thread and high shared memory usage, largely due to inefficient kernel implementations. These factors contribute to the low SMOCC (Fig. 4c).

### 4.2 CHALLENGES AND STRATEGIES FOR CONCURRENT EXECUTION

Concurrent execution on end-user devices leads to interference and resource contention. This section examines how latency-sensitive applications perform under various resource management strategies when all GenAI models fit in GPU memory. Scenarios with larger models are discussed in the Appendix. Since CONSUMERBENCH focuses on profiling a given strategy rather than implementing new ones, we evaluate and report two default and commonly used strategies. Users can add other custom strategies that are workload-aware and better than the generic baselines.

**Greedy Resource Allocation:** This is the default strategy where the kernels of every application greedily occupy GPU resources when they are scheduled, in a first-come-first-serve (FCFS) manner.

- **Performance:** ImageGen performs similarly to how it did when it ran exclusively on the GPU (§4.1). However, both Chatbot and LiveCaptions experience slowdowns. LiveCaptions suffers from starvation and performs particularly poorly, missing SLOs for almost all requests.

---

[2]The model was unable to identify the language used, causing the audio segment to be re-encoded and delayed.

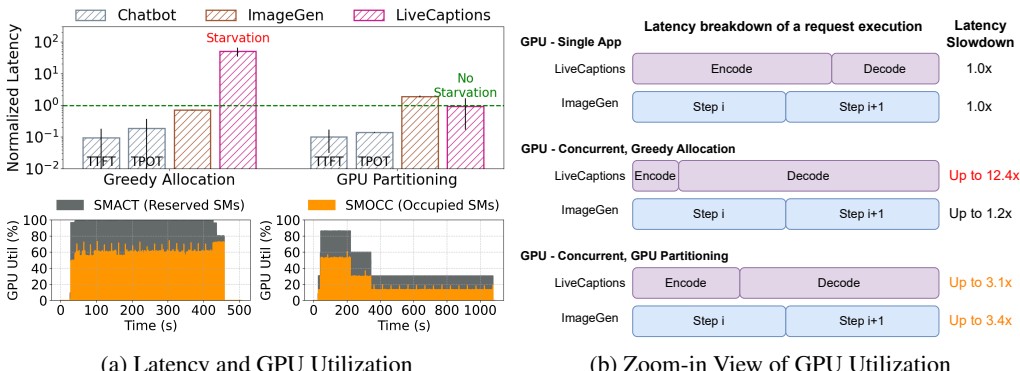

(a) Latency and GPU Utilization    (b) Zoom-in View of GPU Utilization

Figure 5: Application performance & GPU util using greedy resource allocation and GPU partitioning.

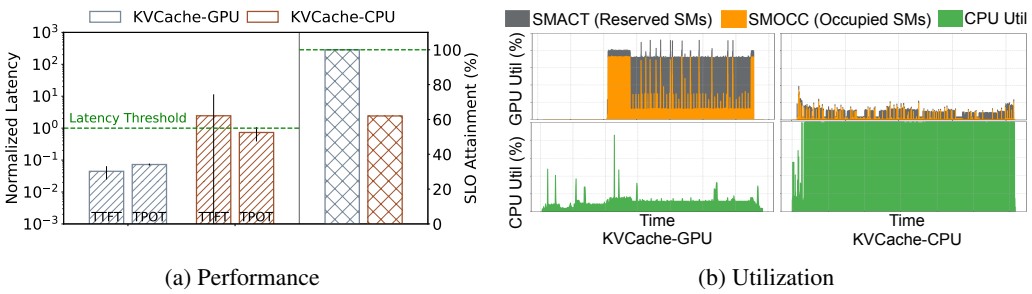

(a) Performance          (b) Utilization

Figure 6: Comparison of Chatbot performance and resource utilization with GPU vs. CPU KV cache.

- **Analysis:** Greedy allocation results in unfair resource reservation, leading to starvation of some applications. A closer look at each request (Fig. 5b) reveals that LiveCaptions' small decode kernels are stalled by ImageGen's large kernels during concurrent execution. As a result, the decode phase in LiveCaptions runs $30\times$ slower compared to exclusive GPU access, leading to a $12.4\times$ increase in the average end-to-end request latency.

**Static GPU Partitioning:** This strategy uses NVIDIA MPS Corporation (2025c) to equally reserve GPU resources (33% of SMs) for each of the three latency-sensitive applications.

- **Performance:** The latency of every application degrades more gracefully compared to greedy allocation, as shown in Fig. 5a. This causes ImageGen to narrowly miss its SLOs, while LiveCaptions is able to meet its SLOs for majority of the requests.
- **Analysis:** GPU partitioning prevents stalling by enforcing equal resource reservation, resulting in predictably higher latencies for all applications (Fig. 5b). However, MPS leads to GPU underutilization by rigidly assigning 33% of GPU cores to each application, even when other partitions are idle. This inflexibility produces a stairstep pattern in SMACT/SMOCC metrics and prevents ImageGen from using available GPU resources after others finish (see GPU utilization in Fig. 5a, right). This causes ImageGen to miss SLOs despite available compute capacity.

### 4.2.1 STATIC MODEL SHARING VIA INFERENCE SERVERS

Given the limited GPU memory on end-user devices, sharing a single foundation model across multiple GenAI applications with similar input-output modalities is a desirable strategy Apple Inc. (2024a). In datacenters, this is typically achieved using inference servers, but such infrastructure is generally lacking on end-user devices. In this section, we evaluate whether inference servers can effectively facilitate model sharing in local environments.

We use the locally-deployed llama.cpp inference server with the Llama-3.2-3B model to serve requests for Chatbot (latency-sensitive task) and DeepResearch (background task). To support DeepResearch's requirement for a large context window, we configure llama.cpp with a 16GB KV cache, matching the model's 128K token context window. To conserve GPU memory, we launch llama.cpp with the `--no-kv-offload` option that stores the KV cache in CPU memory (referred to as Chatbot-

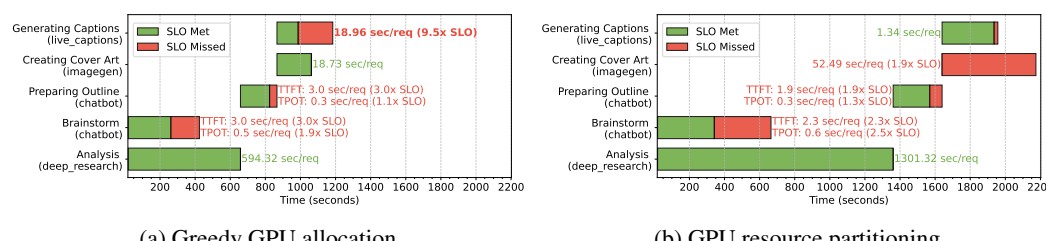

(a) Greedy GPU allocation          (b) GPU resource partitioning

Figure 7: E2E latency & SLO attainment for content-creation workflow w/ and w/o GPU partitioning.

KVCache-CPU). In contrast, running the default Chatbot configuration with a smaller KV cache allows both applications to execute concurrently on the GPU, but forces DeepResearch to use a smaller context window, resulting in degraded output quality. Fig. 6 shows the GPU/CPU utilization and latency for both Chatbot and Chatbot-KVCache-CPU when running alongside DeepResearch.

**Performance & system-level metrics:** Compared to Chatbot, Chatbot-KVCache-CPU exhibits high variance in results and misses its SLO for approximately 40% of its requests. To avoid the overhead of loading the KV cache into the GPU, Chatbot-KVCache-CPU performs attention operations on the CPU, leading to high CPU utilization and correspondingly low GPU utilization.

**Analysis:** Inference servers use static configurations for shared models, which cannot accommodate different performance needs of multiple applications. Configuring a large KV cache for DeepResearch negatively impacts the latency of Chatbot. Hence, naively sharing models via inference servers leads to resource conflicts and suboptimal performance when applications require conflicting settings.

### 4.3 REAL-WORLD USER WORKFLOW

We evaluate end-user experience by simulating a digital content creation workflow with sequential and concurrent tasks: brainstorming (Chatbot-KVCache-CPU), analyzing existing content (Deep-Research), preparing scripts (Chatbot), creating cover image (ImageGen), and generating captions (LiveCaptions). DeepResearch and Chatbot-KVCache-CPU share a single model via llama.cpp to mimic real-world resource constraints. The configuration for this workflow is provided in the Appendix. Fig. 7 shows latencies for workflow tasks with and without GPU partitioning.

**Analysis:** Greedy GPU allocation shortens the end-to-end workflow time by 45% compared to partitioning, mainly due to faster completion of DeepResearch. In contrast, GPU partitioning limits the resources for DeepResearch, causing delays for subsequent tasks. GPU partitioning, on the other hand, avoids stalling of LiveCaptions by reserving 33% of the GPU while gracefully reducing ImageGen performance by 1.8×. Thus, despite better fairness, partitioning increases total workflow runtime, highlighting a trade-off: greedy allocation risks starvation but optimizes utilization, while partitioning ensures fairness at the cost of efficiency.

### 4.4 EVALUATION ON APPLE SILICON

We use CONSUMERBENCH to evaluate our applications on a MacBook M1 Pro Apple Inc. (2024b) laptop with 32GB of unified memory, six performance cores, and two efficiency cores. The detailed results are included in the Appendix due to brevity. Overall, MacBook achieves a better trade-off for GPU utilization and fairness of resources for concurrent applications. However, CONSUMER-BENCH reveals that the computationally intensive kernels of ImageGen and LiveCaptions suffer high slowdowns on MacBook compared to our baseline setup.

## 5 INSIGHTS REVEALED BY CONSUMERBENCH

### 5.1 INSIGHTS FOR BUILDING AND IMPLEMENTING EFFICIENT GENERATIVE AI MODELS

Existing research on GenAI model development for end-user devices primarily focuses on reducing the memory footprint while maintaining generation quality comparable to larger models. However,

CONSUMERBENCH reveals that these methods alone are not sufficient for high system performance (such as response latency) and practical deployments on end-user devices.

**Architectural Efficiency.** The architecture of models should be designed to utilize GPU resources efficiently. This means avoiding practice that can lead to low SM occupancy, such as requiring many intermediate results that demand excessive registers or shared memory per thread. For example, using bounded activation functions constrains the dynamic range of intermediate results, enabling lower-bit representations and reducing shared memory pressure during inference Howard et al. (2019) (§4.1).

**Implementation Awareness in Kernel Design.** The implementation of models needs to be aware of the underlying GPU architecture. This includes creating kernels that leverage architectural features such as warp size and memory hierarchy, as well as optimizing for specialized components such as tensor core utilization (§4.1).

**Concurrency-Aware Kernel Development.** Model implementations should assume concurrent GPU execution by independent applications. Kernels should be developed in a manner that allows the GPU scheduler to achieve both high GPU utilization and fairness in resource allocations across all running applications (§4.2).

## 5.2 Insights for Systems and Inference Frameworks

System and infrastructure developers should provide more flexibility in how models are deployed and how resources are utilized on end-user devices.

**Flexible Resource Management.** Existing GPU partitioning schemes are often static. This can lead to either poor fairness in resource scheduling among competing applications or underutilization of resources. CONSUMERBENCH's experiments reveal a significant need for advancements in system development to achieve dynamicity and flexibility in memory management (§4.2).

**SLO-Aware Scheduling.** Alongside dynamic memory management, there is a need for SLO-aware scheduling of requests specifically for GenAI applications running on end-user devices. This ensures applications meet their user-facing performance targets (§4.2, §4.3).

**Configurable Inference Servers.** Inference servers that support the sharing of a single model among multiple applications need to allow for higher configurability. This is particularly important because applications sharing a model may have different performance goals (§4.2.1). This highlights the need for servers to be more adaptable to varying application SLOs.

## 6 Limitations

CONSUMERBENCH currently assumes that all models are locally deployed on end-user devices and does not address scenarios like edge-cloud collaborative speculative decoding Hao et al. (2024); Oh et al. (2025), where a small local model collaborates with a larger datacenter model. Additionally, while not fundamental to CONSUMERBENCH, the current implementation targets laptops and servers, without support for more constrained devices like mobile phones or sensors.

CONSUMERBENCH does not evaluate GPU partitioning with CUDA Green Context NVIDIA (2025a) because it requires application-level modifications and lacks Python bindings. CONSUMERBENCH does not evaluate GPU partitioning using Multi-Instance GPU (MIG), since it is not supported in consumer-grade GPUs NVIDIA (2025b). We also did not find any existing open-source and transparent dynamic scheduling solution for sharing a single-GPU among multiple applications. This lack of transparent dynamic single-GPU management has also been discussed in other related work Coppock et al. (2025).

## 7 Conclusion

This paper presents CONSUMERBENCH, a comprehensive benchmarking framework for evaluating GenAI applications on end-user devices. It enables users to define realistic workflows and monitors application-level metrics and system performance under concurrent execution. CONSUMERBENCH reveals inefficiencies in resource sharing and kernel scheduling, offering valuable insights to users in developing more efficient models and system optimizations tailored to end-user devices.

## 8 ETHICS STATEMENT

To the best of our knowledge, CONSUMERBENCH does not raise questions regarding the Code of Ethics.

## 9 REPRODUCIBILITY STATEMENT

The implementation of CONSUMERBENCH, along with instructions on how to set it up and reproduce the results in the paper, is uploaded to the Supplementary Material.

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

APPENDICES

# A    ADDITIONAL RESULTS

## A.1    SYSTEM-LEVEL METRICS FOR APPLICATIONS RUNNING EXCLUSIVELY ON GPU

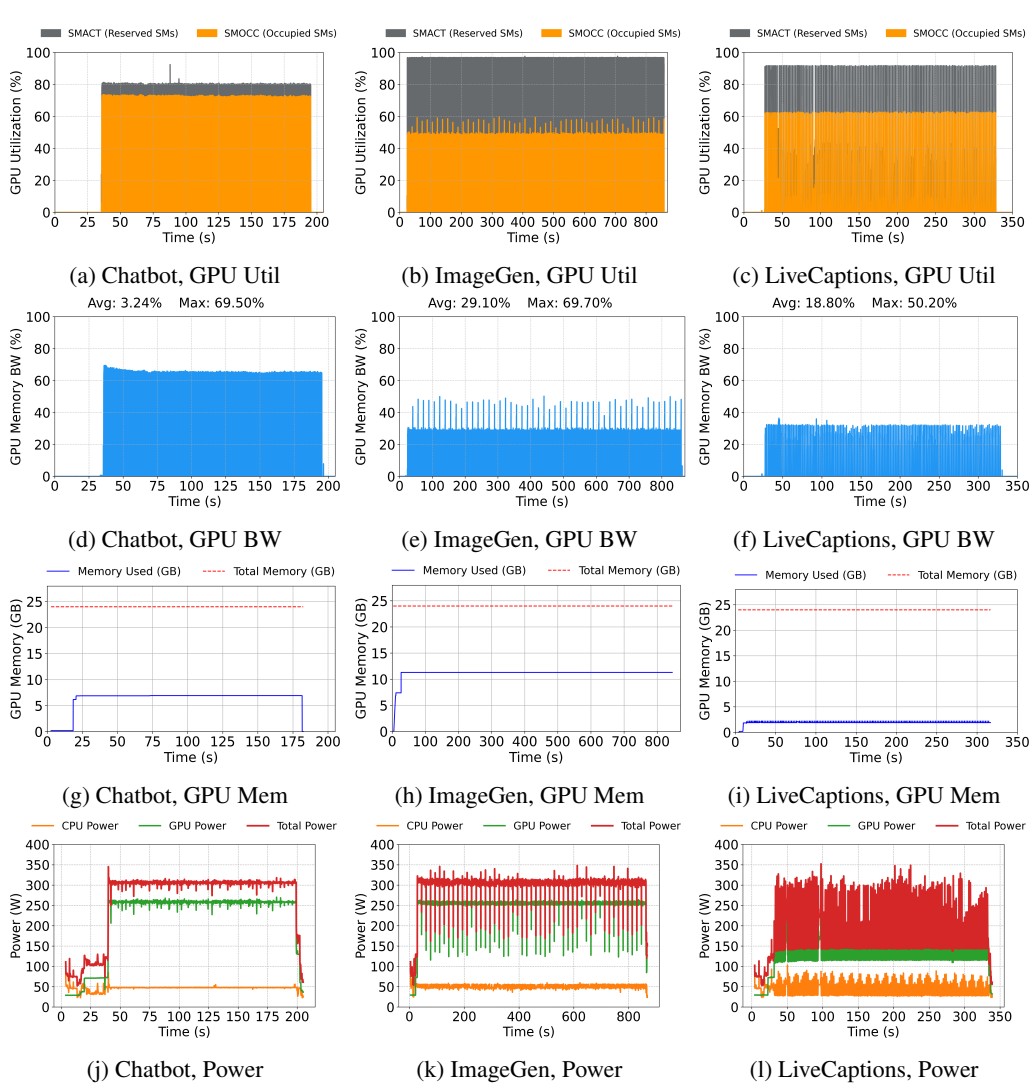

Figure 8: Running applications exclusively on the GPU.

Fig. 8 shows the GPU utilization, memory bandwidth consumption, memory utilization and power consumption when running latency-sensitive applications exclusively on the GPU as reported by CONSUMERBENCH. These are augmented results for Section 4.1 in the paper. Overall, Chatbot consumes the most GPU bandwidth, while ImageGen requires the most amount of GPU memory. All applications have a similar peak power consumption despite the difference in the GPU utilization.

## A.2    SYSTEM-LEVEL METRICS FOR APPLICATIONS RUNNING EXCLUSIVELY ON CPU

Fig. 9 shows the resource utilization when running latency-sensitive applications exclusively on CPU. These are augmented results for Section 4.1 in the paper. The applications are bottlenecked by compute as opposed to memory, evident by the high CPU utilization and low memory bandwidth consumption. Finally, applications require significantly less power when executing on CPU compared to GPU.

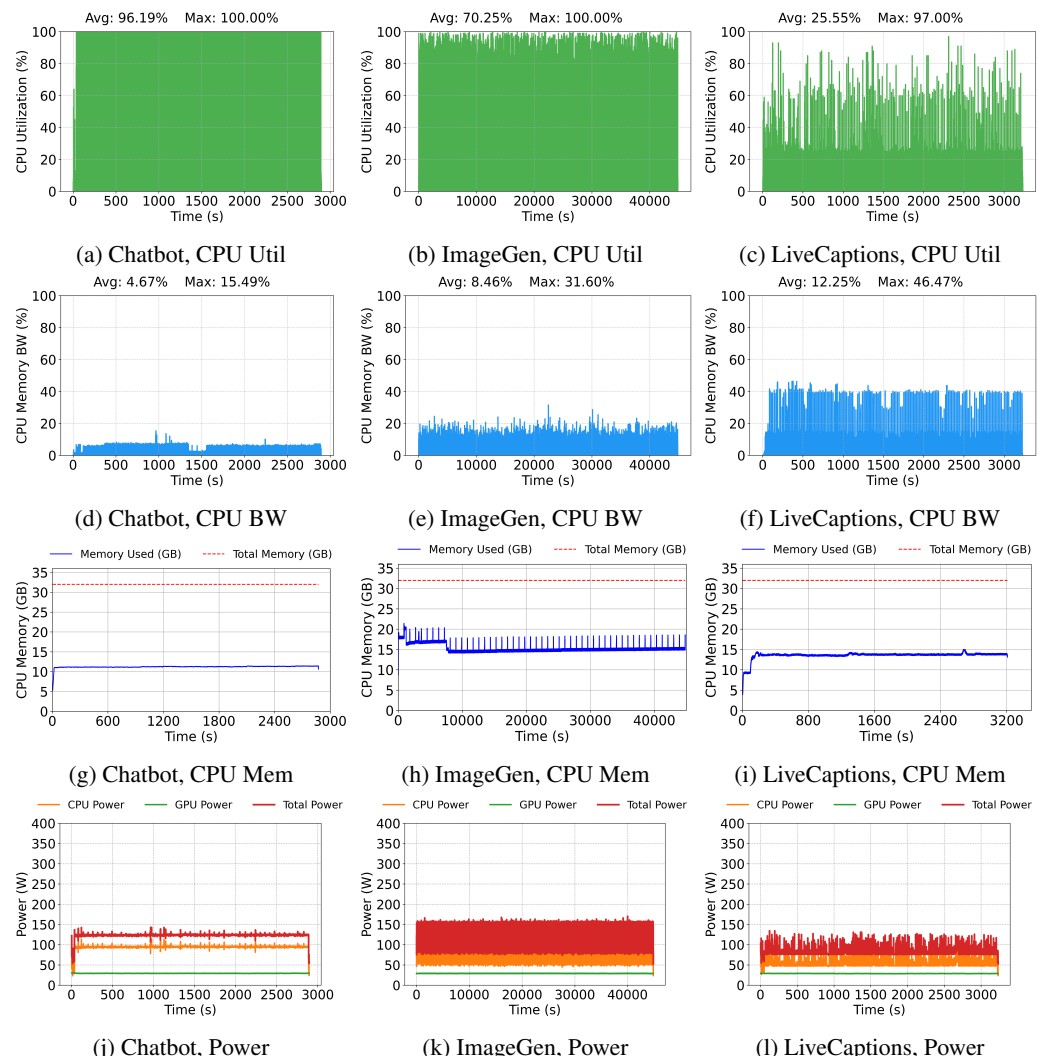

Figure 9: Running applications exclusively on the CPU.

### A.3 SYSTEM-LEVEL METRICS FOR RUNNING CHATBOT WITH GPT-OSS-20B

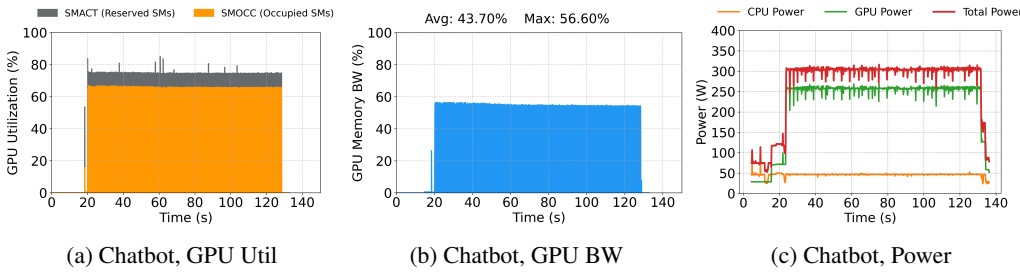

Figure 10: Running Chatbot with GPT-OSS-20B exclusively on the GPU

Fig. 10 and Fig. 11 shows the performance of running Chatbot with GPT-OSS-20B OpenAI et al. (2025) on GPU and CPU exclusively. Similar to running Chatbot with Llama-3.2-3B AI (2024) (see Fig. 8a), SMOCC for GPU utilization is high due to efficient kernel implementation in llama.cpp. When running on CPU exclusively, Chatbot shows high CPU utilization and low CPU bandwidth utilization, similar to findings in Fig. 9.

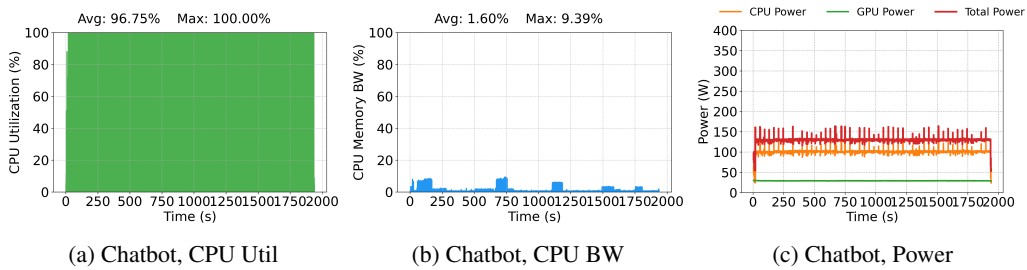

(a) Chatbot, CPU Util     (b) Chatbot, CPU BW     (c) Chatbot, Power

Figure 11: Running Chatbot with GPT-OSS-20B exclusively on the CPU

## A.4 SYSTEM-LEVEL METRICS FOR CONCURRENT EXECUTION OF APPLICATIONS

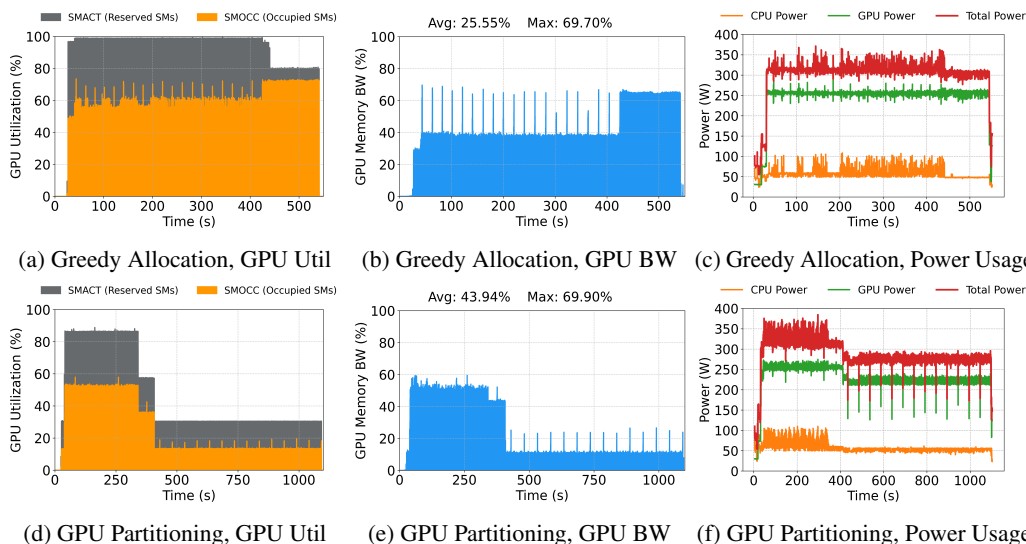

(a) Greedy Allocation, GPU Util   (b) Greedy Allocation, GPU BW   (c) Greedy Allocation, Power Usage

(d) GPU Partitioning, GPU Util   (e) GPU Partitioning, GPU BW   (f) GPU Partitioning, Power Usage

Figure 12: Running applications concurrently on the GPU.

Fig. 12 shows the GPU metrics and power consumption when running Chatbot, ImageGen, and LiveCaptions concurrently on the GPU. These are augmented results for Section 4.2 in the paper. Greedy resource allocation consumes more power on average compared to static GPU partitioning. This directly follows from the fact that static GPU partitioning underutilizes the GPU, as shown in Fig. 12d.

## A.5 CONCURRENTLY EXECUTING APPLICATIONS WITH LARGER MODELS

We use CONSUMERBENCH to evaluate the performance of applications with larger models that do not concurrently fit in the GPU memory of our server. Specifically, we change the model used in Chatbot from Llama-3.2-3B to Llama-3.1-8B that requires 16GB of memory without accounting for the KV Cache. We execute the Chatbot exclusively on CPU while concurrently running ImageGen and LiveCaptions on the GPU. Fig. 13 shows the performance of the applications using greedy GPU allocation and static GPU partitioning. These are augmented results for Section 4.2 in the paper. Fig. 14 and Fig. 15 further show the GPU metrics, CPU metrics and power usage of the different GPU management strategies.

Overall, Chatbot exhibits reduced performance with the larger model compared to Llama-3.2-3B, leading to SLO violations. Although LiveCaptions also experiences SLO violations, its resource starvation is alleviated due to reduced contention when not all three applications share the GPU simultaneously. Lastly, partitioning the GPU between ImageGen and LiveCaptions effectively eliminates starvation for LiveCaptions, though it causes ImageGen to run slightly slower compared to scenarios with greedy resource allocation.

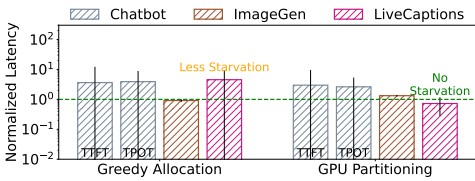

Figure 13: Normalized latency of running larger applications concurrently using greedy allocation and static GPU partitioning

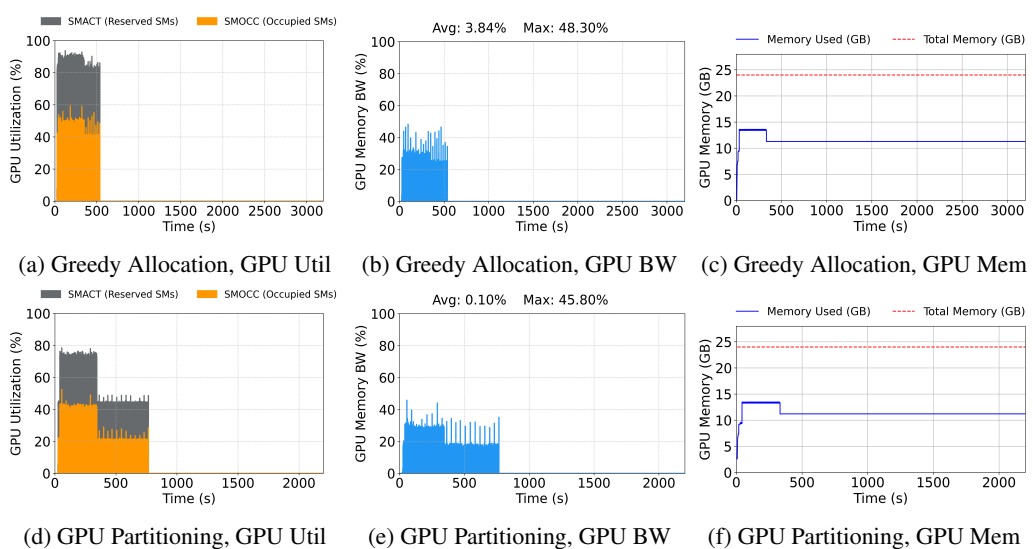

(a) Greedy Allocation, GPU Util  (b) Greedy Allocation, GPU BW  (c) Greedy Allocation, GPU Mem

(d) GPU Partitioning, GPU Util  (e) GPU Partitioning, GPU BW  (f) GPU Partitioning, GPU Mem

Figure 14: GPU metrics of running larger applications concurrently.

### A.6  SYSTEM-LEVEL METRICS FOR STATIC MODEL SHARING VIA INFERENCE SERVERS

Fig. 16 and Fig. 17 show the GPU metrics, CPU metrics, and power usage of running Chatbot with static model sharing via llama.cpp. These are augmented results for Section 4.2.1 in the paper.

Chatbot-KVCache-CPU overall consumes lower GPU resources, but is bottlenecked on the CPU compute due to attention operations on the CPU, evident from the high CPU utilization. Furthermore, performing CPU computations leads to lower power consumption for Chatbot-KVCache-CPU compared to Chatbot.

### A.7  SYSTEM-LEVEL METRICS FOR REAL-WORLD USER WORKFLOW

Fig. 18 and Fig. 19 show the GPU metrics, CPU metrics, and power usage of running the digital content creation workflow using greedy GPU resource allocation and static GPU partitioning. These are augmented results for Section 4.3 in the paper.

Greedy GPU resource allocation overall consumes more GPU resources, and has a higher peak power consumption compared to static GPU partitioning. However, the end-to-end time required for greedy resource allocation is 45% lower than GPU partitioning. This implies that the total power consumed in executing the digital content creation workflow is lower for greedy allocation compared to GPU partitioning.

## B  APPLE SILICON EXPERIMENTS

We conducted all experiments on an Apple MacBook Pro (M1 Pro chip) with 32GB unified memory, 6 performance cores, 2 efficiency cores, and 200 GB/s memory bandwidth. These are augmented results for Section 4.4 in the paper.

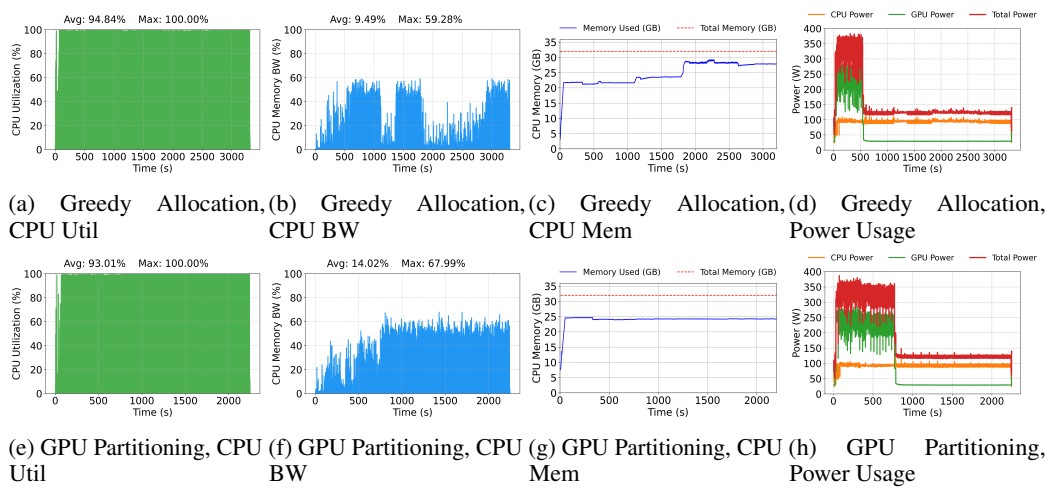

Figure 15: CPU metrics and power usage of running larger applications concurrently.

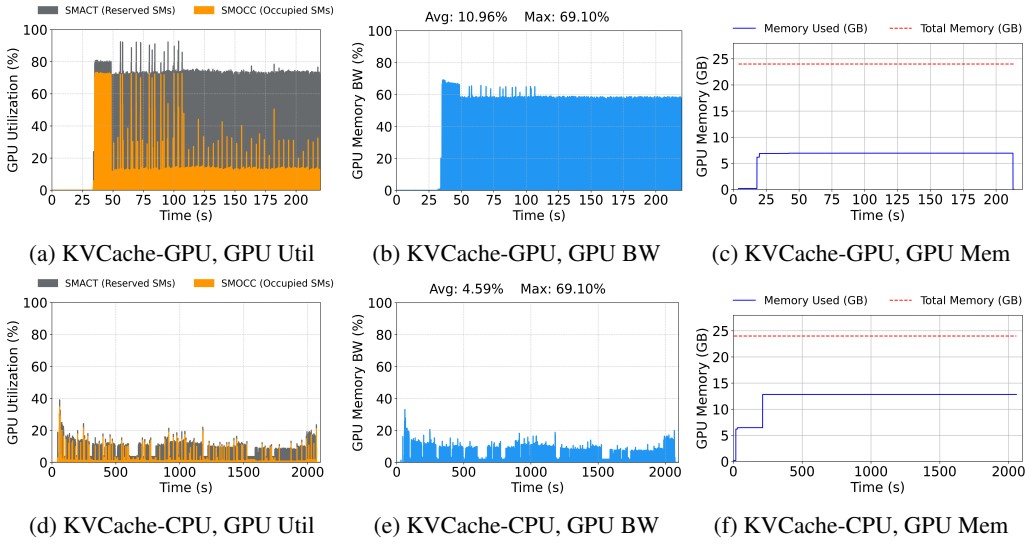

Figure 16: GPU metrics of running applications with static model sharing via inference servers.

**System-level metrics** We use the powermetrics tool of Mac to monitor GPU utilization and power consumption on Apple Silicon devices. Due to the closed-source nature of Apple's hardware, we are unable to report additional metrics such as memory bandwidth utilization. The GPU utilization values provided by powermetrics reflect the proportion of reserved Streaming Multiprocessors (SMs), specifically SMACT. Notably, the power consumption observed on Apple Silicon is substantially lower than that of our previously evaluated Intel server with an NVIDIA GPU, which is expected given the MacBook Pro's lower power capacity as a laptop.

**Application Optimizations.** Our GenAI optimizations for Apple Silicon include:

- **Chatbot & DeepResearch.** We configured the llama.cpp server with the `-metal` flag to optimize GPU kernel execution for Apple Silicon's ARM-based architecture. The Llama-3.2-3B model runs efficiently through this Metal-accelerated backend, handling both conversational and research workloads.

- **ImageGen.** Our local image generation server uses the MPS (Metal Performance Shaders) backend, mirroring llama.cpp's Apple Silicon optimizations. We employ the SD-v1-4 model instead of the NVIDIA-optimized SD-3.5-medium-turbo variant, as it demonstrates better performance on Apple's unified memory architecture.

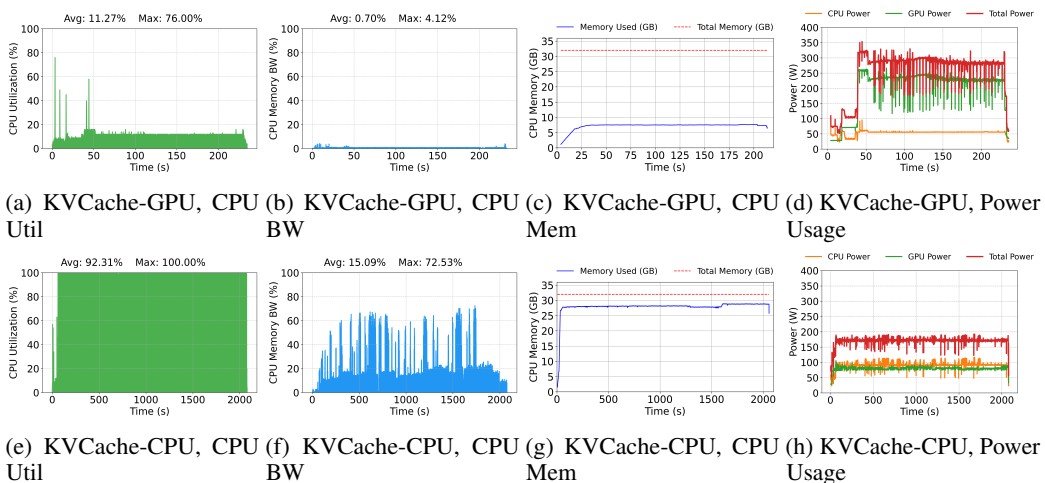

Figure 17: CPU metrics and power usage of running applications with static model sharing via inference servers.

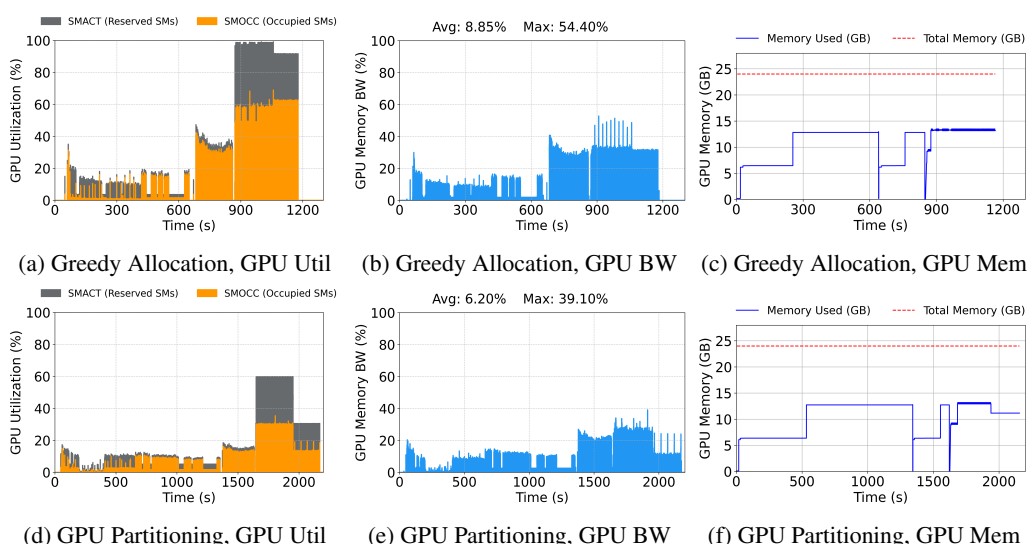

Figure 18: GPU metrics of running the digital content creation workflow.

- **LiveCaptions.** The MLX framework accelerates our Whisper-Large-v3-turbo implementation for real-time audio processing. While maintaining model parity with our default server, we adjusted the service-level objective (SLO) to 4 seconds (from 2 seconds) and modified audio chunking intervals to accommodate Apple Silicon's higher transcription latency.

## B.1 RUNNING APPLICATIONS EXCLUSIVELY VS. CONCURRENTLY

Fig. 20 shows the normalized latency and SLO attainment of the applications when they run exclusively and concurrently on the Apple Silicon. Fig. 21 further shows the GPU utilization and power usage of each scenario. When applications are run in isolation, they are able to meet their SLOs for the majority, if not all, of their requests. However, when applications are executed concurrently, we observe a different pattern compared to the Intel server setup. Specifically, ImageGen experiences a slight performance degradation, while LiveCaptions suffers a significant decline. This behavior suggests that Apple Silicon attempts to fairly schedule GPU compute resources among applications, but the scheduling is suboptimal, leading to resource starvation for LiveCaptions. Additionally, Apple Silicon does not support static GPU partitioning, and we were unable to explore alternative application configurations.

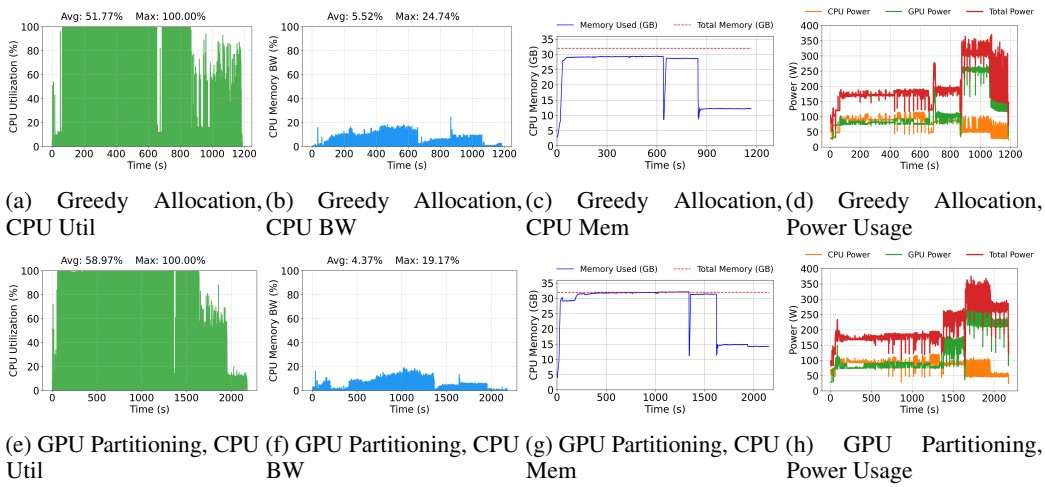

(a) Greedy Allocation, CPU Util    (b) Greedy Allocation, CPU BW    (c) Greedy Allocation, CPU Mem    (d) Greedy Allocation, Power Usage

(e) GPU Partitioning, CPU Util    (f) GPU Partitioning, CPU BW    (g) GPU Partitioning, CPU Mem    (h) GPU Partitioning, Power Usage

Figure 19: CPU metrics and power usage of running the digital content creation workflow.

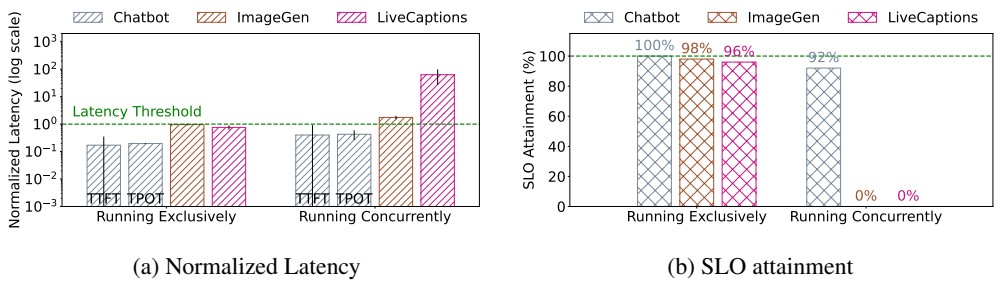

(a) Normalized Latency          (b) SLO attainment

Figure 20: (a) Latencies normalized to SLO requirements and (b) SLO attainment for Chatbot, Image Generation, and Live Caption running exclusively or concurrently on the Apple Silicon.

### B.2 STATIC MODEL SHARING VIA INFERENCE SERVERS

Fig. 22 shows the normalized latency and SLO attainment of running Chatbot and Chatbot-KVCache-CPU on Apple Silicon. Fig. 23 (a,b) shows the GPU utilization, while Fig. 23 (d,e) shows the power usage of each scenario. The performance for Chatbot-KVCache-CPU is similar on Apple Silicon compared with the Intel server. This is for similar reasons. Chatbot-KVCache-CPU does not use GPU cores for attention computation in Chatbot-KVCache-CPU, incurring slowdowns compared to Chatbot.

### B.3 END-TO-END WORKFLOW

We execute the digital content creation workflow on Apple Silicon and present its performance in Fig. 24. Fig. 23c and Fig. 23f shows the GPU utilization and power usage of the workflow. This workflow demonstrates improved fairness in resource allocation on Apple Silicon. Overall, the end-to-end application latency remains comparable to that of the Intel server, while LiveCaptions experiences slightly lesser resource starvation than greedy resource allocation under Intel platform ($8\times$ in Apple Silicon compared to $9.5\times$ in Intel server). Consequently, Apple Silicon achieves a slightly more balanced trade-off between application SLO adherence and overall workflow efficiency compared to the Intel server.

## C CONFIGURATION FOR DIGITAL CONTENT CREATION WORKFLOW

Fig. 25 shows the YAML configuration for the digital content creation workflow, as described in Section 4.3 in the paper.

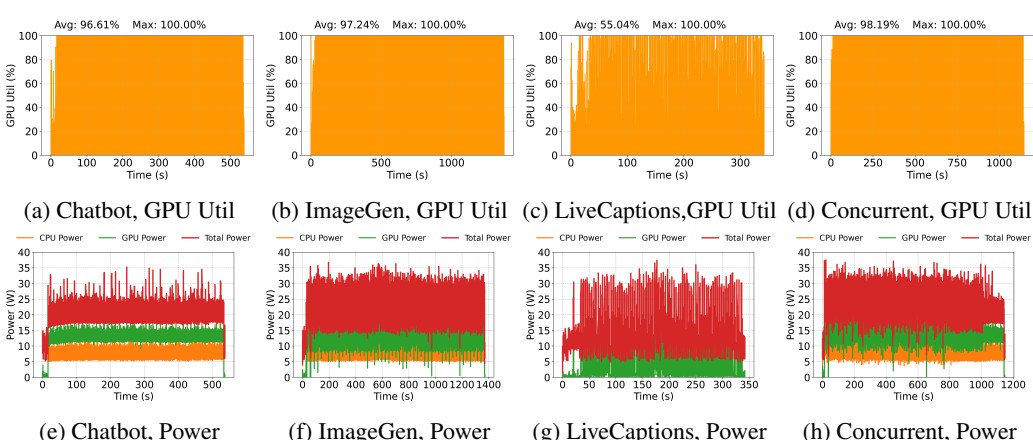

Figure 21: Metrics of running applications exclusively and concurrently on the Apple Silicon.

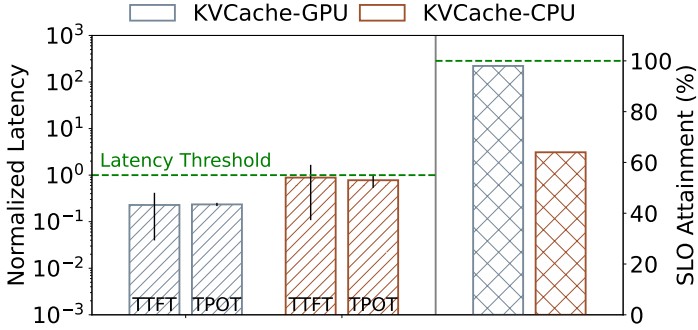

Figure 22: Normalized latency and SLO attainment for Chatbot and Chatbot-KVCache-CPU on the Apple Silicon.

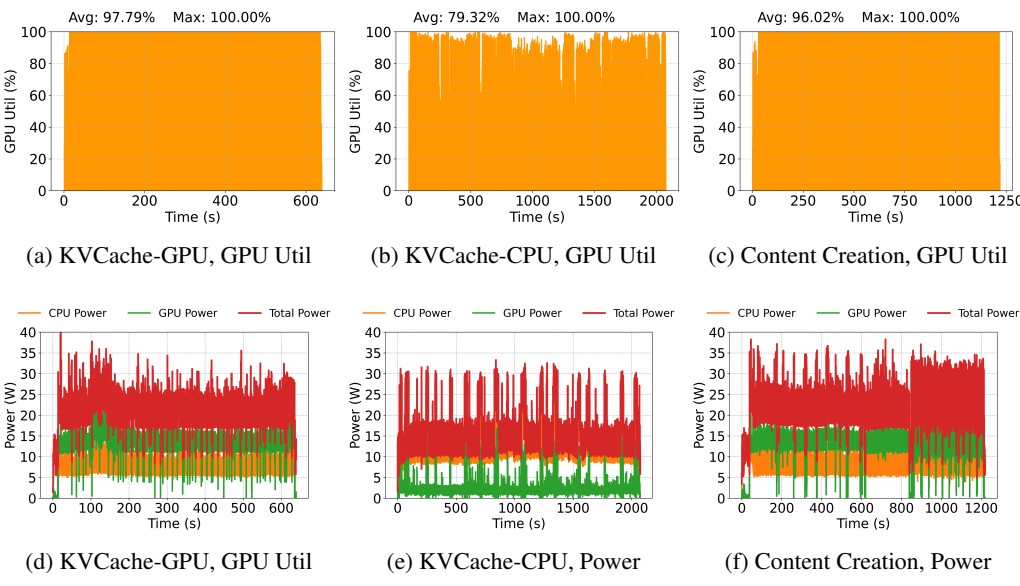

Figure 23: Metrics of running Chatbot, Chatbot-KVCache-CPU and content creation workflow on the Apple Silicon.

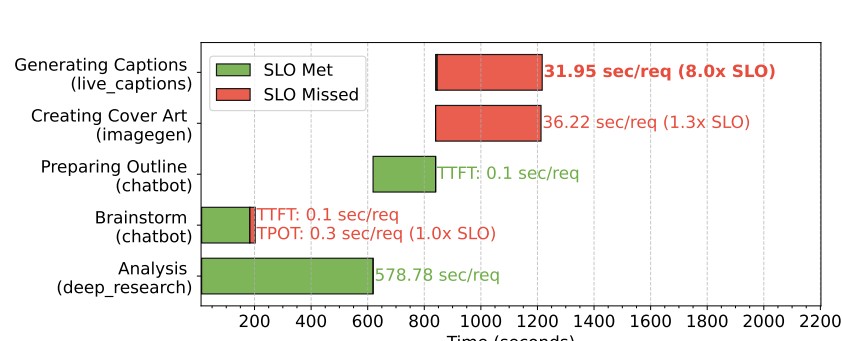

Figure 24: E2E latency & SLO attainment for content-creation workflow running on the Apple Silicon.

```yaml
 1  Brainstorm (chatbot):
 2      model: openai/meta-llama/Llama-3.2-3B-Instruct
 3      num_requests: 10
 4      device: gpu
 5      type: chatbot
 6      mps: 100
 7      slo: [1s, 0.25s]
 8
 9  Analysis (deep_research):
10      model: openai/meta-llama/Llama-3.2-3B-Instruct
11      num_requests: 1
12      device: gpu
13      type: deep_research
14      mps: 100
15
16  Preparing Outline (chatbot):
17      model: openai/meta-llama/Llama-3.2-3B-Instruct
18      num_requests: 20
19      device: gpu
20      type: chatbot
21      mps: 100
22      slo: [1s, 0.25s]
23
24  Creating Cover Art (imagegen):
25      server_model: stable-diffusion-3.5-medium-turbo
26      num_requests: 10
27      device: gpu
28      type: imagegen
29      mps: 100
30      slo: 1s
31
32  Generating Captions (live_captions):
33      num_requests: 1
34      device: gpu
35      type: live_captions
36      mps: 100
37      slo: 2s
38
39  workflows:
40      analysis:
41          uses: Analysis (deep_research)
42          background: true
43
44      brainstorm:
45          uses: Brainstorm (chatbot)
46
47      outline:
48          uses: Preparing Outline (chatbot)
49          depend_on: ["brainstorm", "analysis"]
50
51      cover_art:
52          uses: Creating Cover Art (imagegen)
53          depend_on: ["outline"]
54
55      generate_captions:
56          uses: Generating Captions (live_captions)
57          depend_on: ["outline"]
58
59
```

Figure 25: Full YAML configuration of the content creation workflow.

