# OpenReview forum: "ConsumerBench: Benchmarking Generative AI Applications on End-User Devices"
_ICLR.cc/2026/Conference — Submitted to ICLR 2026_

### Official Review · Reviewer_em9g · 2025-10-19

**Soundness:** 3
**Presentation:** 3
**Contribution:** 3
**Rating:** 6
**Confidence:** 4

**Summary:**

The paper introduces ConsumerBench, a benchmarking framework designed to evaluate the runtime and system efficiency of Generative AI (GenAI) applications running on end-user devices (e.g., laptops, smartphones). Unlike existing benchmarks that assume dedicated hardware, ConsumerBench emulates multi-application, resource-constrained environments, measuring both application-level SLOs (latency, throughput) and system-level metrics (CPU/GPU utilization, memory bandwidth, power).

Experiments on consumer GPUs (RTX 6000) reveal:
1. Greedy GPU allocation causes starvation for lightweight, latency-sensitive applications.
2. Static GPU partitioning improves fairness but lowers utilization.
3. Shared inference servers with fixed configurations can cause conflicting SLO satisfaction.

The authors distill design insights for architecture-aware kernels, SLO-aware scheduling, and configurable inference servers, highlighting gaps in current GenAI runtime systems.

**Strengths:**

1. The paper identifies a critical and underexplored problem: efficient concurrent execution of heterogeneous GenAI workloads on consumer devices.
2. This paper provides an end-to-end benchmarking suite with user-configurable DAG-based workflows, automated metric collection, and extensible APIs for custom apps.
3. This paper is easy to follow. Well-structured experiments reveal non-trivial interactions between applications.
4. The authors provide implementation details, configuration examples, and supplementary materials for replication.

**Weaknesses:**

1. I’m not entirely sure, but NVIDIA MPS might support dynamic resource allocation rather than static partitioning. You can start the NVIDIA MPS and run multiple jobs on the same set of GPUs without additional configuration. I think this could serve as a new baseline.
2. The evaluation only considers greedy and static partitioning, without implementing or comparing against dynamic schedulers (e.g., kernel-level preemption or SLO-prioritized scheduling). This limits the depth of system insights. In fact, several resource allocation papers have explored inference scheduling, for example, Fairness in Serving Large Language Models (OSDI). The authors could consider including additional baselines from this line of work.
3. The study measures latency and utilization but lacks user-centric Quality of Experience (QoE) metrics or perceptual thresholds that better represent real usage.
4. The title formatting and the “Anonymous authors” placeholder in the template look a bit unusual.

**Questions:**

See weaknesses.

---

> ### Author Response · Authors · 2025-11-21
>
> __Concern 1: Additional evaluation with NVIDIA MPS:__
>
> Thank you for your suggestion. We additionally evaluate ConsumerBench on Quadro RTX 6000 GPU with the configuration in Table 1. We compare the setup where 100 MPS is applied to all applications, static GPU partitioning and the greedy resource allocation strategies.  While using 100 MPS helps in reducing the contention, the starvation problem still exists when we run concurrent applications, as specified below. We will add this as a new baseline to our paper.
>
>
> | **Application (SLO)**            | **With MPS All 100** | **Static Partitioning** | **Greedy Allocation** |
> | -------------------------------- | -------------------- | -------------------- | --------------------- |
> | **Chatbot – TTFT (SLO = 1s)**    | 0.092s               | 0.124s               | 0.103s                |
> | **Chatbot – TPOT (SLO = 0.25s)** | 0.022s             | 0.034s               | 0.039s                |
> | **ImageGen (SLO = 1s)**          | 0.687s                   | __1.870s__               | 0.980s                |
> | **LiveCaptions (SLO = 2s)**      | __11.756s__            | __2.436s__              | __28.916s__               |

---

> > ### Author Response · Authors · 2025-11-21
> >
> > __Concern 2: Evaluating with dynamic resource scheduling strategies:__
> >
> > We have tried evaluating various dynamic scheduling policies, as specified below. However, none of the scheduling policies work in practice for unmodified applications.
> >
> > - CUDA Priority-Streams: We evaluated priority-streams and found that it is only effective to manage priorities within a single context (i.e., single application) but not across contexts or applications.
> > - CUDA Green Context: Similar to priority-streams, only works within a single context.
> > - REEF [OSDI'22]: Only supports a limited number of DNN models.
> > - TGS [NSDI'23]: Only supports 2 applications concurrently with predefined binary priority. Also, we tried running TGS and found it does not run on newer models and frameworks such as diffusers (used in ImageGen), and faster-whisper (used in LiveCaptions).
> > - Fairness in Serving Language Models [OSDI'24] : Only supports LLM workloads implemented on top of LightLLM, not arbitrary GenAI applications.
> > - Orion [EuroSys'24]: Requires an offline profiling stage to gather kernel profiles, needing application-level modifications. The scheduler only supports DNN model kernels.
> > - LithOS [SOSP'25]: Not open-sourced.
> > - Torpor [ATC'25]: Torpor is designed for serverless environments, where its primary objective is to optimize the loading of idle models for on-demand inference in cluster deployments. Torpor executes only one serverless function on a GPU at a time and disallows concurrent workloads.
> >
> > We additionally simulate an "oracle" adaptive GPU scheduling policy. We perform a grid search on the optimal MPS configuration for each application with the goal of maximizing SLO attainment. We use the concurrent workload of Chatbot, ImageGen and LiveCaptions with the model setup in Table 1 on the RTX6k node.
> >
> > We first perform a grid search for the MPS threshold for each application running standalone and pick the smallest threshold that can still satisfy the SLO constraint. We then run the optimal MPS configuration (“With MPS Optimal”) and compare with greedy allocation and with default 100 MPS to all applications (“With MPS All 100”).
> >
> > The table below shows the results. With the optimal MPS configuration, all applications manage to attain their SLOs, demonstrating that an effective adaptive scheduling policy could resolve resource contention between applications.
> >
> > | **Application (SLO)**            | **With MPS Optimal** | **With MPS All 100** | **Greedy Allocation** |
> > | -------------------------------- | -------------------- | -------------------- | --------------------- |
> > | **Chatbot – TTFT (SLO = 1s)**    | 0.154s               | 0.092s               | 0.103s                |
> > | **Chatbot – TPOT (SLO = 0.25s)** | 0.039s               | 0.022s               | 0.039s                |
> > | **ImageGen (SLO = 1s)**          | 0.927s               | 0.687s               | 0.980s                |
> > | **LiveCaptions (SLO = 2s)**      | 1.25s                | __11.756s__              | __28.916s__               |

---

> > > ### Author Response · Authors · 2025-11-21
> > >
> > > __Concern 3: Enabling user-centric Quality of Experience (QoE) metrics:__
> > >
> > > ConsumerBench allows users to specify custom metrics per application, simply in the YAML configuration, and evaluates the application performance against the set metrics. While we measure utilization, latency, bandwidth and power, users can specify QoE metrics such as token-delivery-timeline for LLMs [1], real-time-factor for Automatic Speech Recognition [2], and so on.
> > >
> > > [1] Liu, J., Wu, Z., Chung, J., Lai, F., Lee, M., & Chowdhury, M. (2024). Andes: Defining and Enhancing Quality-of-Experience in LLM-Based Text Streaming Services. ArXiv, abs/2404.16283.
> > >
> > > [2] C. Arriaga, A. Pozo, J. Conde and A. Alonso, "Assessing Latency in ASR Systems: A Methodological Perspective for Real-Time Use.," in IEEE Internet Computing, doi: 10.1109/MIC.2025.3614363.
> > >
> > > __Concern 4: Formatting Issues:__
> > >
> > > We have checked the format of the paper and fixed issues in the title and the “Anonymous authors” placeholder. We will upload a revised version of the paper to reflect the changes.

---

### Official Review · Reviewer_9PWZ · 2025-10-27

**Soundness:** 2
**Presentation:** 3
**Contribution:** 3
**Rating:** 4
**Confidence:** 3

**Summary:**

This paper presents ConsumerBench, a benchmarking framework for evaluating Generative AI applications running concurrently on end-user devices. Users specify applications, models, service-level objectives (SLOs), and inter-application dependencies in a YAML configuration file, which the system compiles into a directed acyclic graph (DAG) of application instances. ConsumerBench then executes and measures these workloads to expose inefficiencies in resource management, GPU sharing, and SLO-aware scheduling under multi-application concurrency.

**Strengths:**

1. Novel concurrency focus: The paper addresses multi-application inference, a relatively unexplored yet practically important problem for end-user AI systems.
2. The experimental results yield clear, practical takeaways for developers seeking to improve performance and fairness when multiple generative AI applications share hardware resources.
3. The use of YAML-based configuration and the DAG-based task execution model make the framework easily extensible — new applications, models, or metrics can be integrated with minimal effort.

**Weaknesses:**

1. Despite claiming a focus on “end-user devices,” all experiments are performed on a single workstation with an RTX 6000 GPU. Evaluations on consumer-class GPUs (e.g., RTX 4060/4070) or integrated accelerators would strengthen the paper’s external validity.
2. Each task uses a fixed model configuration. Demonstrating results across multiple models per modality would better validate that the benchmark’s findings generalize beyond specific architectures.
3. The paper discusses SLO-aware resource allocation but does not evaluate any dynamic or adaptive scheduling strategies. This weakens the central argument and leaves open the question of how ConsumerBench would perform under more optimal schedulers.

**Questions:**

1. Have you considered expanding the benchmark to include more task types and custom metrics, allowing users to choose flexible workload combinations for evaluation?
2. What was the reasoning behind using only one hardware setup? Would you consider evaluating on more representative consumer devices or cross-vendor platforms to demonstrate generality?

---

> ### Author Response · Authors · 2025-11-21
>
> __Concern 1: Running on more consumer-class GPUs and models:__
>
> - Thank you for your suggestions. Below are some highlight results in ConsumerBench on different consumer-grade GPUs, software platforms and different AI models per modality (results for Apple Silicon are provided in Figure 20 in the Appendix). Note that ConsumerBench is extensible to support any combination of model, hardware setup, and OS configurations. We will also revise our paper with the new results as well as system-level metrics not shown here due to format constraints.
>
>
> | **Config** | **Hardware Setup** | **OS**        | **GPU Memory** | **Model-Chatbot**       | **Model-ImageGen**           | **Model-LiveCaption**                        |
> | ---------- | ------------------ | ------------- | ---------------- | ----------------------- | ---------------------------- | -------------------------------------------- |
> | **A**      | RTX 4080           | Windows WSL   | 16GB             | Gemma-2B          | SD-3-medium            | Whisper-large-v3-turbo             |
> | **B**      | RTX 3060           | Windows WSL | 12GB             | Llama-3.2-1B       | SD-3.5-medium-Q8        | Whisper-small                  |
> | **C**      | Jetson Orin Nano   | Ubuntu        | 8GB              | Llama-3.2-1B Q8_0   |        -                     | Whisper-tiny |
> | **D**      | RTX6000            | Ubuntu        | 24GB             | Llama-3.2-3B            | SD-3.5-medium-turbo             | Whisper-large-v3-turbo              |
> | **E**      | RTX6000            | Ubuntu        | 24GB             | Qwen2.5-7B              | kandinsky-2-2-decoder | Whisper-medium                   |
> | **F**      | RTX6000            | Ubuntu        | 24GB             | GPT-OSS-20B             | kandinsky-2-2-decoder  | Whisper-small                    |
>
> Due to a lack of support for MPS in WSL and Jetson Orin Nano, we evaluate each configuration under greedy allocation strategies. Below, we show the comparison of SLO attainment for each application using a workflow that runs three applications concurrently.
>
> __Takeaway__: We observe that in all these scenarios, the problem identified in Figure 5(a) of the paper still persists: LiveCaptions is consistently stalled when concurrently executing alongside other applications on the GPU. This shows that our findings from ConsumerBench translate to different models, hardware and OSs.
>
> | **Hardware Setup** | **Chatbot TTFT**(SLO = 1s) | **Chatbot TPOT**(SLO = 0.25s) | **ImageGen**(SLO = 1s) | **LiveCaption** (SLO = 2s) |
> | ------------------ | ------------------------------ | --------------------------------- | -------------------------- | ----------------------------- |
> | **A**              | 0.070                          | 0.026                             | 0.621                      | __23.242__                        |
> | **B**              | 0.062                          | 0.026                             | __1.651__                      | __91.504__                        |
> | **C**              | 0.097                          | 0.031                             | -                     | __45.488__                        |
> | **D**              | 0.100                          | 0.039                             | 0.874                      | __40.274__                        |
> | **E**              | 0.060                          | 0.017                             | 0.126                      | __6.217__                         |
> | **F**              | 0.098                          | 0.011                             | 0.121                      | __15.479__                        |

---

> ### Author Response · Authors · 2025-11-21
>
> __Concern 2: Evaluating with dynamic resource scheduling strategies:__
>
> We have tried evaluating various dynamic scheduling policies, as specified below. However, none of the scheduling policies work in practice for unmodified applications.
>
> - CUDA Priority-Streams: We evaluated priority-streams and found that it is only effective to manage priorities within a single context (i.e., single application) but not across contexts or applications.
> - CUDA Green Context: Similar to priority-streams, only works within a single context.
> - REEF [OSDI'22]: Only supports a limited number of DNN models.
> - TGS [NSDI'23]: Only supports 2 applications concurrently with predefined binary priority. Also, we tried running TGS and found it does not run on newer models and frameworks such as diffusers (used in ImageGen), and faster-whisper (used in LiveCaptions).
> - Fairness in Serving Language Models [OSDI'24] : Only supports LLM workloads implemented on top of LightLLM, not arbitrary GenAI applications.
> - Orion [EuroSys'24]: Requires an offline profiling stage to gather kernel profiles, needing application-level modifications. The scheduler only supports DNN model kernels.
> - LithOS [SOSP'25]: Not open-sourced.
> - Torpor [ATC'25]: Torpor is designed for serverless environments, where its primary objective is to optimize the loading of idle models for on-demand inference in cluster deployments. Torpor executes only one serverless function on a GPU at a time and disallows concurrent workloads.
>
> We additionally simulate an "oracle" adaptive GPU scheduling policy. We perform a grid search on the optimal MPS configuration for each application with the goal of maximizing SLO attainment. We use the concurrent workload of Chatbot, ImageGen and LiveCaptions with the model setup in Table 1 on the RTX6k node.
>
> We first perform a grid search for the MPS threshold for each application running standalone and pick the smallest threshold that can still satisfy the SLO constraint. We then run the optimal MPS configuration (“With MPS Optimal”) and compare with greedy allocation and with default 100 MPS to all applications (“With MPS All 100”).
>
> The table below shows the results. With the optimal MPS configuration, all applications manage to attain their SLOs, demonstrating that an effective adaptive scheduling policy could resolve resource contention between applications.
>
> | **Application (SLO)**            | **With MPS Optimal** | **With MPS All 100** | **Greedy Allocation** |
> | -------------------------------- | -------------------- | -------------------- | --------------------- |
> | **Chatbot – TTFT (SLO = 1s)**    | 0.154s               | 0.092s               | 0.103s                |
> | **Chatbot – TPOT (SLO = 0.25s)** | 0.039s               | 0.022s               | 0.039s                |
> | **ImageGen (SLO = 1s)**          | 0.927s               | 0.687s               | 0.980s                |
> | **LiveCaptions (SLO = 2s)**      | 1.25s                | __11.756s__              | __28.916s__               |

---

> > ### Author Response · Authors · 2025-11-21
> >
> > __Concern 3: Enabling custom metrics and adding new tasks:__
> >
> > ConsumerBench is extensible and allows users to specify custom metrics per application, simply in the YAML configuration, and allows the application interface to evaluate the SLO violation (such as memory/power limits, or other custom metrics).
> > Adding new tasks in ConsumerBench is trivial and only requires defining some task-specific functions such as `setup()`, `execute()` and `cleanup()` (please see Section 3.3 of the paper for more details).
> > ConsumerBench also allows users to specify any flexible workload combinations and use different SLO levels for different combinations.

---

### Official Review · Reviewer_rMux · 2025-10-28

**Soundness:** 3
**Presentation:** 3
**Contribution:** 3
**Rating:** 4
**Confidence:** 3

**Summary:**

This paper presents a novel benchmark called ConsumerBench, which helps evaluates performances of multi-application scenarios executed concurrently on constrained hardware. This is a useful benchmark given the intensive requirements of applications these days. The paper also showcases an example that helped provide potential insights that can be drawn from using the benchmark.

**Strengths:**

The paper identifies an useful, practical gap in benchmarking applications under constrained on-device resources. It also presents careful thought out workflow and analysis, along with an example usecase where the use of such a benchmark/framework might help bring more insights on the concurrent execution and where the limits might occur. The findings are well written and presented with clarity.

**Weaknesses:**

The paper provides a benchmark that is useful in terms of software aspects for applications running concurrently on constrained resources. However, it hasn't mentioned any consideration for hardware impacts, which can further influence the performance of the applications on the end-user devices. Furthermore, the paper could explain more on the usecases and usefulness of the benchmark, such as how the workflow/findings scale and provide systematic insights across different architecture and types of end-user devices.

**Questions:**

- Can this benchmark be used in a container or sandbox setting, such as to simulate and understand the performance of target applications on a device before actual implementation/purchase? The main usecase seems to be for an existing device.
- The paper identified several factors affecting performance. To what extent can the bottleneck and processes identified in the analysis be used to better schedule the applications?

---

> ### Author Response · Authors · 2025-11-21
>
> __Concern 1: Running on containerized environments and evaluating on different hardware:__
>
> - Thank you for the suggestion. ConsumerBench is designed to be modular, and we have containerized it to be deployed for different hardware platforms. In order to illustrate this, we report the results of ConsumerBench on different GPUs and Operating Systems below (results for Apple Silicon are provided in Figure 20 in the Appendix). Note that ConsumerBench is extensible to support any combination of model, hardware setup, and OS configurations. We will also revise our paper with the new results as well as system-level metrics not shown here due to format constraints.
>
>
> | **Config** | **Hardware Setup** | **OS**        | **GPU Memory** | **Model-Chatbot**       | **Model-ImageGen**           | **Model-LiveCaption**                        |
> | ---------- | ------------------ | ------------- | ---------------- | ----------------------- | ---------------------------- | -------------------------------------------- |
> | **A**      | RTX 4080           | Windows WSL   | 16GB             | Gemma-2B          | SD-3-medium            | Whisper-large-v3-turbo             |
> | **B**      | RTX 3060           | Windows WSL | 12GB             | Llama-3.2-1B       | SD-3.5-medium-Q8        | Whisper-small                  |
> | **C**      | Jetson Orin Nano   | Ubuntu        | 8GB              | Llama-3.2-1B Q8_0   |        -                     | Whisper-tiny |
> | **D**      | RTX6000            | Ubuntu        | 24GB             | Llama-3.2-3B            | SD-3.5-medium-turbo             | Whisper-large-v3-turbo              |
> | **E**      | RTX6000            | Ubuntu        | 24GB             | Qwen2.5-7B              | kandinsky-2-2-decoder | Whisper-medium                   |
> | **F**      | RTX6000            | Ubuntu        | 24GB             | GPT-OSS-20B             | kandinsky-2-2-decoder  | Whisper-small                    |
>
> Due to a lack of support for MPS in WSL and Jetson Orin Nano, we evaluate each configuration under greedy allocation strategies. Below, we show the comparison of SLO attainment for each application using a workflow that runs three applications concurrently.
>
> __Takeaway__: We observe that in all these scenarios, the problem identified in Figure 5(a) of the paper still persists: LiveCaptions is consistently stalled when concurrently executing alongside other applications on the GPU. This shows that our findings from ConsumerBench translate to different models, hardware and OSs.
>
> | **Hardware Setup** | **Chatbot TTFT**(SLO = 1s) | **Chatbot TPOT**(SLO = 0.25s) | **ImageGen**(SLO = 1s) | **LiveCaption** (SLO = 2s) |
> | ------------------ | ------------------------------ | --------------------------------- | -------------------------- | ----------------------------- |
> | **A**              | 0.070                          | 0.026                             | 0.621                      | __23.242__                        |
> | **B**              | 0.062                          | 0.026                             | __1.651__                      | __91.504__                        |
> | **C**              | 0.097                          | 0.031                             | -                     | __45.488__                        |
> | **D**              | 0.100                          | 0.039                             | 0.874                      | __40.274__                        |
> | **E**              | 0.060                          | 0.017                             | 0.126                      | __6.217__                         |
> | **F**              | 0.098                          | 0.011                             | 0.121                      | __15.479__                        |

---

> > ### Author Response · Authors · 2025-11-21
> >
> > __Concern 2: Scheduling applications based on the analysis in ConsumerBench:__
> >
> > The analysis derived from ConsumerBench can be used to develop OS-like GPU/CPU schedulers that are able to efficiently share GPU resources for concurrent applications. Below are two different and composable ways in which applications can be scheduled based on the analysis of ConsumerBench.
> >
> > - __Application scheduling based on resource usage of AI models:__ The analysis in ConsumerBench shows that different applications exhibit different GPU compute and memory utilization, when it comes to allocating resources, and using the resources efficiently (Figure 4). For example, the ImageGen application allocates all the GPU resources when running, and uses long-running and larger kernels, which stalls other concurrent applications such as LiveCaptions with smaller kernels. This motivates scheduling policies that are able to study the GPU behavior of AI applications and accordingly spatially share GPU resources such that there is no stalling, or there are pre-emption mechanisms to disallow ImageGen from stalling other applications.
> >
> > - __Request-level scheduling based on SLOs:__ Different applications also exhibit different SLO requirements. For example, Chatbot is a real-time application with low latency SLOs, while DeepResearch is a background and long-running application. When they share the same LLM, the Chatbot requests get stalled due to DeepResearch requests (Section 4.2.1). This analysis in ConsumerBench reveals that GPU resources should be scheduled in an SLO-aware manner at a request granularity rather than an application granularity.

---

> > > ### Comment · Reviewer_rMux · 2025-11-24
> > >
> > > Dear authors,
> > >
> > > Thank you for your clarification on different GPUs/OS and discussion on scheduling applications. I will keep the current score which acknowledge the contributions of the manuscript but also that there are scope to further enhance it in terms of distinction with other benchmarks, more detailed analysis to demonstrate the usecases across different architecture and analysis of hardware impacts.

---

### Official Review · Reviewer_pqy1 · 2025-11-01

**Soundness:** 2
**Presentation:** 3
**Contribution:** 1
**Rating:** 2
**Confidence:** 4

**Summary:**

The paper introduces consumerbench, a benchmarking application for four GenAI workloads. The benckmark is used to benchmark the performance of the four applications on a 2018 RTX600 GPU, with an Intel Xeon Gold 6126 CPU (2.60GHz, 24 cores) and 32GB of system memory (DRAM). The results show that three out of the four applications tested, achieve their required SLOs, with the third one not achieving the slo for 1.5% of the audio samples.

**Strengths:**

Benchmarking hardware is an important problem. This paper aims to improve the current State-of-the-art of edge/small device benchmarking by benchmarking workflows of tasks.

**Weaknesses:**

My main issue is with the novelty and depth of the work. For example, as a benchmark, when comparing the suggested benchmark with PalmBench (cited in the paper), the authors have a much more limited set of applications/Models. When it comes to insights from the experiments, the results are very well known. For example, the KTransofrmer project (open-sourced with a paper in SOSP 2025) has been setup to solve many of the insights discussed.

Another issue, for a benchmarking paper, one typically needs to run on many configurations. Going back to the PalmBench paper, they test with three operating systems on nine different hardware platforms. Testing on only two platforms does not allow for a good enough evaluation.


Writing:
1- The paper almost exclusively cites Arxiv papers/versions of the paper with a handful of exceptions. Please fix this!

**Questions:**

1. Can you expand your experiments to more models, scenarios, and more hardware/OS configurations?
2. Besides the wokflow, what else is different from PalmBench?

---

> ### Author Response · Authors · 2025-11-21
>
> __Concern 1: How is ConsumerBench different from PalmBench?__
>
> ConsumerBench fundamentally differs from PalmBench in the following ways:
> - PalmBench evaluates the efficiency and accuracy of single model deployment on mobile devices, while ConsumerBench evaluates the efficiency for concurrent execution of GenAI applications on end-user devices with realistic user workflows.
> - PalmBench focuses solely on LLM evaluation, while ConsumerBench evaluates applications across multiple modalities, including text and image generation and audio processing.
> - The insights from PalmBench focus on efficient LLM deployments on mobile devices in a single-model single-device setting, while those from ConsumerBench focus more broadly on GPU and CPU resource management, request scheduling for inference frameworks, and efficient AI model architectures.

---

> > ### Author Response · Authors · 2025-11-21
> >
> > __Concern 2: Can KTransformer solve many of the insights proposed in ConsumerBench?__
> >
> > KTransformer optimizes single MoE models using heterogeneous CPU/GPU expert placement and specialized kernels. In contrast, ConsumerBench targets concurrent and competing applications running on end-user devices with diverse SLOs. Consequently, KTransformer represents a specific, single-model subset within the broader ConsumerBench scope.
> >
> > __Insights derived from ConsumerBench:__
> >
> > KTransformer does not fully address the insights and suggestions specified in ConsumerBench.
> > - __Flexible Resource Management:__ ConsumerBench advocates for a global OS-like scheduler to dynamically schedule and allocate resources on heterogeneous hardware across competing applications, whereas KTransformer statically partitions the MoE model across CPU and GPU without considering other concurrently running applications.
> > - __SLO-Aware Scheduling:__ ConsumerBench advocates for request-level SLO-aware scheduling of GPU resources, when multiple applications with different SLOs share the same LLM (e.g., low-latency Chatbot requests and background DeepResearch requests). KTransformer does not perform SLO-level scheduling of requests to CPU/GPU resources, even though some requests may require low-latency and others may tolerate high latency.
> > - __Configurable Inference Servers:__ ConsumerBench advocates for higher configurability (such as request-specific KV-cache offload) in inference servers to enable sharing GenAI models across different applications with competing goals. KTransformer does not provide higher configurability at the inference server level to enable sharing a single MoE across multiple independent applications.
> >
> > __Evaluating KTransformer:__
> >
> > We run KTransformer on DeepSeek-V2-Lite family of LLMs and report its performance when running in isolation and in  a concurrent setting with LiveCaptions and Imagegen. We find that in the concurrent case, KTransformer incurs a slow down in TTFT as well as TPOT.
> >
> >
> > | **Model Setting**                    | **TTFT** (SLO=1s) | **TPOT** (SLO=0.25s) |
> > | ------------------------------------ | -------- | -------- |
> > | DeepSeek-V2-Lite Q4 Standalone | 0.205s   | 0.025s   |
> > | DeepSeek-V2-Lite Q4 Concurrent | __1.035s__  | 0.142s   |
> > | DeepSeek-V2-Lite Q8 Standalone | 0.719s   | 0.037s   |
> > | DeepSeek-V2-Lite Q8 Concurrent | __1.180s__       | 0.161s        |

---

> > > ### Author Response · Authors · 2025-11-21
> > >
> > > __Concern 3: Can you expand your experiments to more models, scenarios, and hardware/OS configurations?__
> > >
> > > Below are our evaluations on more models and hardware/OS configurations. Note that ConsumerBench is extensible to support any combination of model, hardware setup, and OS configurations.
> > >
> > > | **Config** | **Hardware Setup** | **OS**        | **GPU Memory** | **Model-Chatbot**       | **Model-ImageGen**           | **Model-LiveCaption**                        |
> > > | ---------- | ------------------ | ------------- | ---------------- | ----------------------- | ---------------------------- | -------------------------------------------- |
> > > | **A**      | RTX 4080           | Windows WSL   | 16GB             | Gemma-2B          | SD-3-medium            | Whisper-large-v3-turbo             |
> > > | **B**      | RTX 3060           | Windows WSL | 12GB             | Llama-3.2-1B       | SD-3.5-medium-Q8        | Whisper-small                  |
> > > | **C**      | Jetson Orin Nano   | Ubuntu        | 8GB              | Llama-3.2-1B Q8_0   |        -                     | Whisper-tiny |
> > > | **D**      | RTX6000            | Ubuntu        | 24GB             | Llama-3.2-3B            | SD-3.5-medium-turbo             | Whisper-large-v3-turbo              |
> > > | **E**      | RTX6000            | Ubuntu        | 24GB             | Qwen2.5-7B              | kandinsky-2-2-decoder | Whisper-medium                   |
> > > | **F**      | RTX6000            | Ubuntu        | 24GB             | GPT-OSS-20B             | kandinsky-2-2-decoder  | Whisper-small                    |
> > >
> > > Due to a lack of support for MPS in WSL and Jetson Orin Nano, we evaluate each configuration under greedy allocation strategies. Below, we show the comparison of SLO attainment for each application using a workflow that runs three applications concurrently.
> > >
> > > __Takeaway__: We observe that in all these scenarios, the problem identified in Figure 5(a) of the paper still persists: LiveCaptions is consistently stalled when concurrently executing alongside other applications on the GPU. This shows that our findings from ConsumerBench translate to different models, hardware and OSs.
> > >
> > > | **Hardware Setup** | **Chatbot TTFT**(SLO = 1s) | **Chatbot TPOT**(SLO = 0.25s) | **ImageGen**(SLO = 1s) | **LiveCaption** (SLO = 2s) |
> > > | ------------------ | ------------------------------ | --------------------------------- | -------------------------- | ----------------------------- |
> > > | **A**              | 0.070                          | 0.026                             | 0.621                      | __23.242__                        |
> > > | **B**              | 0.062                          | 0.026                             | __1.651__                      | __91.504__                        |
> > > | **C**              | 0.097                          | 0.031                             | -                     | __45.488__                        |
> > > | **D**              | 0.100                          | 0.039                             | 0.874                      | __40.274__                        |
> > > | **E**              | 0.060                          | 0.017                             | 0.126                      | __6.217__                         |
> > > | **F**              | 0.098                          | 0.011                             | 0.121                      | __15.479__                        |

---

> > > > ### Author Response · Authors · 2025-11-21
> > > >
> > > > __Concern 4: The paper almost exclusively cites arXiv papers/versions of the paper with a handful of exceptions__
> > > >
> > > > We have re-checked all the references and changed the arXiv citations for the papers that have been accepted to a peer-reviewed journal or conference. We will upload the revised paper to reflect the changes.

---

> ### Comment · Reviewer_pqy1 · 2025-11-23
>
> Thanks you for your comments.
> I would like to follow up on your comment. I do not fully understand how you view the difference between ConsumerBench and PalmBench are. I am probably missing something, so this is just a follow up:
>
> 1. While PalmBench does not include multiple modalities for their testing, what is the fundamental difference that comes from images? Why would it matter? I could not really see what is the insight that is gained from ConsumerBench that makes using SD-3.5 or SD-1.4 any special? With the exception of this single line "Spends most of its time in the denoising phase, which uses an attention-based U-Net Oktay et al. (2018). PyTorch’s generic attention kernel used by SD-3.5-Medium-Turbo requires over 150 registers per thread, limiting the number of threads that can run concurrently on each SM. This reduces SMOCC and leads to suboptimal GPU utilization" which is based on running the model exclusively.
>
> 2. When I look at the workflow results, I am still not sure what do we learn from ConsumerBench, that we do not know already today. I also do not see how this would scale beyond some toy examples. To have a good workflow based benchmark, we need to have pre-constructed scenarios that make sense, see e.g., the DeathStarBench Microservice benchmarks. Here you have a single workflow that you evaluate with. From an engineering perspective/novelty perspective, this is less that what one would expect for a paper to be accepted at ICLR in my opinion. From a results/findings perspective, how would any of your results generalize?
>
> 3. Given the above two points, I still fail to see your main contributions.

---

> > ### Author Response · Authors · 2025-12-04
> >
> > __1. Comparison with PalmBench:__
> >
> > ConsumerBench supports different modalities of GenAI models compared to PalmBench. The fundamental difference due to the different modalities lies in kernel diversity and the resulting resource contention patterns. While PalmBench effectively evaluates LLMs in isolation, LLMs exhibit relatively homogeneous runtime behaviors (shown in the table below). In contrast, introducing different modalities (like ImageGen and LiveCaptions) introduces drastically different kernel characteristics (e.g., register pressure, thread occupancy). These differences are the root cause of severe system inefficiencies such as starvation which manifest during concurrent execution—a scenario PalmBench does not cover.
> >
> > As shown in the table below, running different LLM architectures (e.g., Llama-3.2-3B vs. GPT-OSS-20B) exhibits very similar system-level behaviors. They generally maintain high SM Occupancy (SMOCC) relative to their reservation (SMACT), indicating efficient GPU utilization.
> >
> > However, ConsumerBench applications exhibit diverse inefficiencies. As noted in the paper (Fig. 4), ImageGen suffers from low SM occupancy due to high register pressure (U-Net attention kernels), while LiveCaptions exhibits variable occupancy.
> >
> > | Application | Model | SMACT (Reserved SMs) | SMOCC (Occupied SMs) | Efficiency Gap |
> > | :--- | :--- | :--- | :--- | :--- |
> > | Chatbot (LLM) | Llama-3.2-3B | ~80.8% | ~72.4% | Low (Efficient) |
> > | Chatbot (LLM) | GPT-OSS-20B | ~77.8% | ~65.2% | Low (Efficient) |
> > | ImageGen | SD-3.5-Turbo | ~90% | ~45% | High (Register Bound) |
> > | LiveCaptions | Whisper-v3 | ~90% | ~60% | Mixed (Kernel Dependent) |
> >
> > -----------------
> > __2. Importance of Workflows in ConsumerBench:__
> >
> > ConsumerBench is fundamentally a benchmarking framework designed to enable users to define and execute their own custom workflows. The primary contribution of this work is the framework that allows users to test realistic scenarios spanning multiple GenAI applications for performing end-to-end tasks. The content-creation workflow is presented in the paper as an example of the scenarios that can be easily created in ConsumerBench. We are open to expanding the suite of pre-constructed scenarios similar to DeathStarBench. To support the organic generation of realistic scenarios, we are developing a leaderboard where users can contribute their own workflows, such as coding assistants or video generation pipelines, using ConsumerBench, based on their actual usage experiences.
> >
> > __Takeaways that are possible using Workflows rather than simple concurrent tests:__
> >
> > Unlike simple concurrent tests in other GPU scheduling works where unrelated AI models are run concurrently, ConsumerBench workflows enforce dependencies (e.g., DeepResearch must finish before Chatbot summarizes). This reveals how resource starvation in an early background task propagates to delay the entire pipeline, which is a compounding effect that static concurrency benchmarks miss. Furthermore, we find that different GenAI applications have significantly different behaviors seen via GPU utilization, and shown in the table above. ConsumerBench helps in consciously running the different applications in a realistic manner (such as a background deep-research task running concurrently with a foreground chatbot or imagegen) rather than arbitrarily executing applications that a user would not run in parallel in the real-world (which is evaluated in other GPU scheduling works).

---

> ### Comment · Reviewer_pqy1 · 2025-11-23
>
> I think I have not made my point clear probably in my review. My point is that, the insights you provide are well known and are already being addressed by State-of-the-Art solutions in MLSys research, e.g., KTransformers, which provides flexible resource management between the CPU and the GPU. This was just one example, however, within the systems community, there has been quite a few papers on each of the insights you bring. For example, the insights revealed by ConsumerBench "for building and implementing efficient GenAI models" are the reason why many people are working on Kernel optimizations and fusion both in industsy and academia. Hence, "KernelBench: Can LLMs Write Efficient GPU Kernels?" in ICML 2024 tried to automate the process to be able to cover the shortcomings idenitfied in 5.1.
>
> When it comes to Section 5.2, the three insights, starting with the first insight are actually not in anyway special for LLMs and GenAI. These are problems that have been studied in the systems community for the past 7 years roughly. KTransformers is just one example project that aims to improve the CPU-GPU performance of LLMs in an OS like fashion. They go a few steps further, in my opinion, in including the CPU, which I believe is the only way that you can run these incredibly heavy workloads on an edge/consumer device.
>
> I truly appreciate that you have run the experiments with KTransformers, but the point, I believe, still stands.

---

> > ### Author Response · Authors · 2025-12-04
> >
> > __Novel insights derived from ConsumerBench analysis:__
> >
> > Our analysis in ConsumerBench provides novel insights that go beyond general efficiency improvements. We show that different applications exhibit distinct discrepancies between SM reservation (SMACT) and SM utilization (SMOCC) ratios. This specific mismatch leads to severe starvation during concurrent execution, a unique observation that standard efficiency benchmarks overlook. This leads to the targeted insight that GPU kernels for end-user devices must be designed such that SMOCC remains generally high to ensure efficient GPU utilization. This observation is more specific and actionable for system designers than simply implementing efficient GenAI models.
> >
> > ConsumerBench also reveals a critical scheduling failure when background and foreground applications share a model—for instance, DeepResearch and Chatbot utilizing the same LLM. Our analysis demonstrates that the background application effectively starves the foreground application, even in the presence of standard batching mechanisms. This observation leads to the specific insight that GPU scheduling must be implemented at a "request-level" rather than an "application-level". This is necessary to ensure that low-latency requests receive higher priority than background requests, even when they share the same inference server.
> >
> > Finally, ConsumerBench highlights the value of building dynamic scheduling strategies that are practically deployable on consumer hardware. While there has been extensive work on dynamic GPU scheduling in recent years, we found that many of these strategies are not practically usable for end-user devices running unmodified GenAI applications on a single shared GPU. Our analysis underscores the gap between theoretical research and deployable solutions. In our next answer, we provide a table detailing different static and dynamic GPU scheduling strategies proposed over the last few years to illustrate their applicability and limitations in this specific context.

---

> ### Comment · Reviewer_pqy1 · 2025-11-23
>
> While I truly appreciate the runs on the new hardware, the shortcomings of MPS are well documented. They are the strawman against almost all GPU sharing papers compare to. The question is however, what else can we learn from ConsumerBench when running on different Hardware platforms, that we do not know. In addition, the assumption that when running a workload, such as the one presented, one would simply split the allocation on the three applications equally is I believe not correct. Most people, if they would ever use MPS at all, would profile the three applications, and then increase the allocation to the ones which requires more resources. This would still be inflexible, however, as said, this is the bulk of work on GPU sharing over the past 6-7 years.

---

> ### Author Response · Authors · 2025-12-04
>
> __Evaluation of ConsumerBench with different GPU scheduling strategies:__
>
> While it would have been ideal to run more dynamic GPU scheduling systems in ConsumerBench, we found that they do not fully support the applications and scenarios considered in our framework. Below we compare and discuss several recently proposed dynamic GPU scheduling solutions as well as other native solutions.
>
> | System | Transparently support arbitrary GenAI models on a single GPU | Resource Scheduling across applications (not within an application) | Support for arbitrary number of concurrent applications | Evaluated |
> | :--- | :--- | :--- | :--- | :--- |
> | **TGS** | Y | Y | N (only 1 High-Priority App) | Y (with just two concurrent apps) |
> | **Orion** | N | Y | Y | N (Does not support newer GenAI models) |
> | **REEF** | N | Y | Y | N (Does not support newer GenAI models) |
> | **MPS/MIG** | Y | Y | Y | Y |
> | **FaaSwap / Torpor** | N | Y | Y | N (Does not support GPU sharing) |
> | **GreenContext** | Y | N | N | N (Does not support concurrent apps) |
> | **Priority-streams** | Y | N | N | N (Does not support concurrent apps) |
> | **KTransformers, llama.cpp** | N | N | N | Y |
> | **AppleSilicon** | Y | Y | Y | Y |
>
> __Results on TGS:__
>
> In the below table we further present the evaluation of TGS when concurrently running ImageGen and LiveCaptions applications.
>
> | Configuration & Priority | ImageGen Performance | LiveCaptions Performance | Observation |
> | :--- | :--- | :--- | :--- |
> | **ImageGen (Low Priority) + LiveCaptions (High Priority)** | Severe SLO Violation | Meets SLO | TGS severely throttles the low priority image generation task to allow real-time audio processing, causing prohibitive slowdown for ImageGen. |
> | **ImageGen (Low Priority) + LiveCaptions (Low Priority)** | Meets SLO | Starvation | Under equal priority, heavy image generation kernels block the GPU, causing total starvation for the lightweight LiveCaptions task. |

---

### Comment · Area_Chair_3UYh · 2025-11-23
**Action Needed: Review Rebuttal and Update Evaluation**

Dear Reviewers,

Thank you, as always, for your valuable contributions and efforts. The authors have now submitted their rebuttal. Please take a moment to review it and provide any necessary follow-up actions, such as additional questions, clarification requests, or updates to your review.

Since the initial ratings ranged from 2 to 6, I kindly ask you to pay close attention to the perspectives of the other reviewers when preparing your final response.

Thank you again for your support.

---

### Meta-Review · Area_Chair_d3uC · 2026-01-06

**Summary:**

This paper presents ConsumerBench, a benchmark where multiple and diverse GenAI applications are ran concurrently on a single end-user device. These application have different workloads and different SLOs that present different systems challenges. The authors benchmark existing strategies and show that no existing solutions are able to handle the workloads in their proposed benchmark.

There were two primary concerns raised by reviewers that were not sufficiently addressed in the benchmark. The first concerns the novelty of the benchmark and the insights generated from it. The second concerns that lack of benchmarking of existing adaptive/dynamic allocation strategies.

The response by the authors to both these concerns is that their benchmark has many concurrent applications and a diverse set of applications. This makes it different from existing benchmarks and make existing allocation strategies inapplicable. I do not believe this argument would have alleviated the reviewers concerns.

Specifically, it is unclear what makes ConsumerBench different from other benchmarks gives new insights or is just different. One possible way to show this would be to make a best effort attempt to benchmark ConsumerBench-like scenarios in a context that current allocation strategies can handle. If the authors could show that the design decisions made for existing methods are wrong for the more complex scenarios in ConsumerBench, that would demonstrate the value of the benchmark.

**Reviewer Concerns:**

## Reviewer pqy1

- Novelty of the benchmark, especially compared to PalmBench. I believe this concern is still outstanding.
- The novelty of the experimental insights. I believe this concern is still outstanding. It is unclear
- Breadth of the benchmark configurations. I believe this concern has been addressed.
- Evaluation with different GPU scheduling strategies. I believe this concern is still outstanding.

## Reviewer rMux

- Containerization or sandboxing for evaluation. I believe this concern is still outstanding. Have some ability to emulate different GPUs (within the same architecture, or GPUs that use the same chip but with different bins) would make reproducibility much easier.

## Reviewer 9PWZ

- Breadth of hardware platforms used for benchmarking.
- Only one model configuration per task.
- Lack of dynamic of adaptive scheduling strategies. I believe this concern is still outstanding.

## Reviewer em9g

- Dynamic resource allocation with nvidia MPS. I believe this concern has been addressed.
- Lack of adaptive/dynamic scheduling strategies. I believe this concern is still outstanding.

**Reviewer Scores:**

I do not believe reviewer scores would have meaningfully changed after rebuttal.

---

### Decision · Program_Chairs · 2026-01-26

Reject